# Cryo-EM structures and functional properties of CALHM channels of the human placenta

**Katarzyna Drożdżyk[1†], Marta Sawicka[1†*], Maria-Isabel Bahamonde-Santos[1], Zaugg Jonas[2], Dawid Deneka[1], Christiane Albrecht[2], Raimund Dutzler[1*]**

[1]Department of Biochemistry, University of Zurich, Zurich, Switzerland; [2]Institute of Biochemistry and Molecular Medicine, University of Bern, Bern, Switzerland

**Abstract** The transport of substances across the placenta is essential for the development of the fetus. Here, we were interested in the role of channels of the calcium homeostasis modulator (CALHM) family in the human placenta. By transcript analysis, we found the paralogs CALHM2, 4, and 6 to be highly expressed in this organ and upregulated during trophoblast differentiation. Based on electrophysiology, we observed that activation of these paralogs differs from the voltage- and calcium-gated channel CALHM1. Cryo-EM structures of CALHM4 display decameric and undecameric assemblies with large cylindrical pore, while in CALHM6 a conformational change has converted the pore shape into a conus that narrows at the intracellular side, thus describing distinct functional states of the channel. The pore geometry alters the distribution of lipids, which occupy the cylindrical pore of CALHM4 in a bilayer-like arrangement whereas they have redistributed in the conical pore of CALHM6 with potential functional consequences.

**\*For correspondence:**
m.sawicka@bioc.uzh.ch (MS);
dutzler@bioc.uzh.ch (RD)

[†]These authors contributed equally to this work

**Competing interests:** The authors declare that no competing interests exist.

## Introduction

The placenta, a complex organ that develops during pregnancy, serves as a hub for the exchange of nutrients and waste products between the mother and the fetus. The flow of substances across the placenta is controlled by two distinct cell-layers, namely fetal capillary endothelial cells and syncytio-trophoblasts (STB). While endothelial cells lining the fetal vessels allow to a certain extent diffusion of (small) molecules through paracellular pathways (*Edwards et al., 1993*; *Lewis et al., 2013*), the STB layer constitutes a tight diffusion barrier. STB are polarized multinucleated cells presenting their apical side towards the maternal blood and their basal side towards the fetal capillaries. Hence, the targeted expression of receptors, channels or transport proteins of the STB determines the directionality of transport across the placenta, which is essential for the adequate development of the fetus (*Lager and Powell, 2012*). Among the transport proteins in the placenta, the role of the calcium homeostasis modulator (CALHM) family is currently unknown. In humans, the CALHM family encompasses six paralogs, some of which function as non-selective channels that are permeable to large substances such as ATP (*Ma et al., 2016*). CALHM1, the currently best characterized paralog, forms ion channels that are activated by depolarization and a decrease in the extracellular $Ca^{2+}$-concentration (*Ma et al., 2012*; *Taruno et al., 2013b*) with mutations being associated with an increased risk for late-onset Alzheimer's disease (*Dreses-Werringloer et al., 2008*). Although functional on its own (*Ma et al., 2012*), in a physiological context CALHM1 probably forms heteromers with the subunit CALHM3 (*Ma et al., 2018b*). These heteromeric channels play an important role as secondary receptors for sweet, bitter and umami taste reception in type II taste bud cells by catalyzing the non-vesicular release of ATP (*Ma et al., 2018b*; *Taruno et al., 2013a*; *Taruno et al., 2013b*). A second paralog with an assigned physiological role is CALHM2, which is ubiquitously expressed and was proposed to mediate ATP release in astrocytes (*Ma et al., 2018a*). The role of other

paralogs has thus far remained elusive. Insight into the structural properties of the family was recently provided from studies on CALHM1 and 2 by cryo-electron microscopy (*Choi et al., 2019*; *Demura et al., 2020*; *Syrjanen et al., 2020*). CALHM2 was found to assemble into undecameric channels, which in certain cases dimerize via contacts on the extracellular side in an assembly that resembles gap-junctions. Based on this architecture, a potential role of CALHM2 as intercellular channels was proposed. Although predicted to be related to connexins and volume-regulated anion channels (VRACs) of the LRRC8 family (*Siebert et al., 2013*), the four membrane-spanning helices of the CALHM subunit exhibit a distinct arrangement, which refutes a common structural ancestry between these protein families (*Choi et al., 2019*). In contrast to CALHM2 channels, the structure of CALHM1 shows a smaller octameric organization (*Demura et al., 2020*; *Syrjanen et al., 2020*), thus suggesting that the functional properties of activation and conduction might not be conserved within the family.

In the present study we were investigating the structural and functional properties of CALHM channels in the context of the placenta. We identified the three paralogs, CALHM2, 4 and 6, to be highly expressed in this organ. A systematic comparative functional characterization by electrophysiology did not reveal pronounced activity of either of these paralogs under conditions where CALHM1 channels are open, thus suggesting that the former might be regulated by different and currently unknown mechanisms. The structural characterization of the three paralogs reveals insight into their organization and into potential gating transitions which, although related to previously described properties of CALHM2, show distinct features with respect to subunit organization, conformational changes and the distribution of lipids residing inside the wide pore of CALHM channels.

## Results

### Expression of CALHM channels in the human placenta

To investigate the role of CALHM channels in the placenta, we have characterized the expression of family members in healthy human placental tissues obtained from term pregnancies. Quantification of transcripts by reverse transcription (RT) PCR revealed high levels of CALHM2, 4, and 6, but comparably low expression of other paralogs (*Figure 1A*). We also analyzed the expression patterns of CALHM paralogs in primary trophoblast cells isolated from healthy term placentae and examined

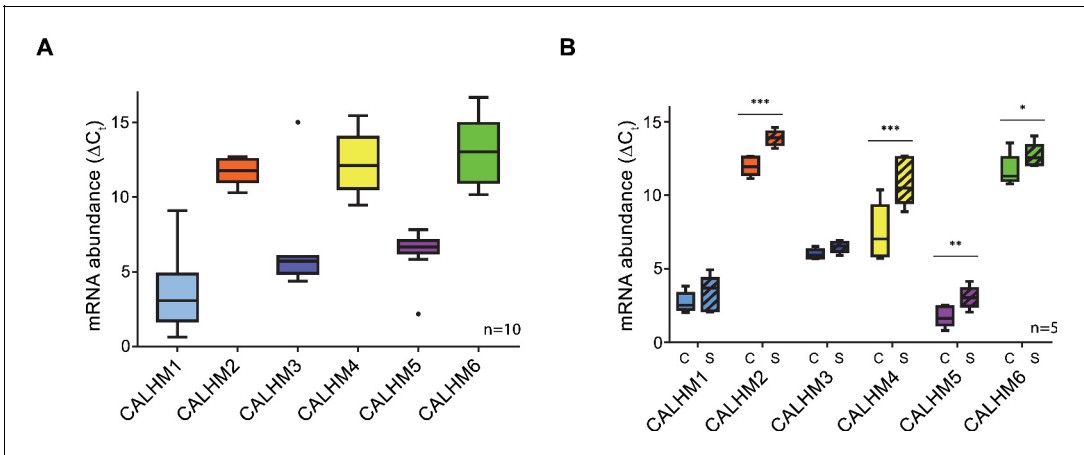

**Figure 1.** Expression analysis of CALHM genes in the human placenta. mRNA abundances of different CALMH paralogs in (**A**), human placental tissues (n = 10) and (**B**), human trophoblasts isolated from healthy term placentae (n = 5) was assessed by quantitative RT PCR and normalized to the reference gene YWHAZ. The relative amounts of the different CALHM genes are shown as $\Delta C_t$ values ($\Delta C_t = C_t$ value of YWHAZ – $C_t$ value of CALHM gene). Comparative transcript data are presented as mRNA abundance ($\Delta Ct = Ct_{reference\ gene} – Ct_{traget\ gene}$) between undifferentiated cytotrophoblasts (C) and differentiated syncytiotrophoblast (S) cells. Data analysis and statistical evaluations were performed using paired 2-way ANOVA with Sidak's multiple comparisons test; *$p<0.05$; **$p<0.01$; ***$p<0.001$.

The online version of this article includes the following figure supplement(s) for figure 1:

**Figure supplement 1.** Comparative mRNA expression data for other physiologically relevant transporters/receptors in the human placenta.

changes during the differentiation process of trophoblast precursor cells into mature STB. We found marked upregulation of CALHM2, 4 and 6 during differentiation, which was most pronounced for CALHM4 (*Figure 1B*) and which might indicate a differentiation-dependent role of these proteins in the human placenta. In all cases, the concentrations of the respective CALHM mRNAs are high in comparison to other transport proteins (*Figure 1—figure supplement 1*) suggesting their involvement in important placenta-related membrane transport processes.

## Functional characterization of CALHM paralogs

Due to the abundance of CALHM2, 4, and 6 in the placenta, we have focused our subsequent investigations on these three paralogs. Apart from CALHM2, which was described to form ATP-conducting channels in astrocytes (*Ma et al., 2018a*), they have so far not been characterized. For a functional comparison of channels composed of the three placental subunits with the well-studied protein CALHM1, we have expressed the homomeric proteins in *X. laevis* oocytes and recorded currents by two-electrode voltage-clamp electrophysiology. This method has previously allowed a detailed characterization of CALHM1, which is activated by depolarization and the removal of $Ca^{2+}$ from the extracellular side (*Ma et al., 2012*). To detect proteins in the plasma membrane, we have used a surface-biotinylation approach and found all placental paralogs to be expressed and targeted at high levels to the surface of oocytes at the time of the measurement (*i.e.* 40–60 hr after injection of mRNA) (*Figure 2A*). Assuming that CALHM proteins form channels of large conductance, we thus can directly compare the average magnitude of recorded currents between populations of oocytes expressing the respective constructs and relate their activation properties in response to voltage change and the depletion of extracellular $Ca^{2+}$ to the well-characterized CALHM1. In our studies, we were able to reproduce the functional hallmarks of CALHM1, which are manifested in the absence of currents at negative voltage and slowly activating currents at positive voltage at millimolar concentrations of extracellular $Ca^{2+}$ and a strong increase of currents upon $Ca^{2+}$ removal in the entire voltage range (*Figure 2*, *Figure 2—figure supplement 1A and B*). In contrast to CALHM1, the current response of homomeric CALHM2, 4, and 6 channels was generally small and within the range of endogenous currents of *X. laevis* oocytes not expressing any of the proteins (*Figure 2A*, *Figure 2—figure supplement 1C-F*). These currents neither showed pronounced voltage-dependence nor were they altered by $Ca^{2+}$-removal in a statistically significant manner (*Figure 2B–D*). We thus conclude that the CALHM channels expressed in the placenta are not regulated in a similar manner as CALHM1-subunit containing channels and that their activation instead proceeds by distinct, currently unknown mechanisms.

## Biochemical characterization and structure determination of CALHM2, 4, and 6

For a biochemical and structural characterization of placental CALHM channels, we have expressed constructs coding for human CALHM2, 4 and 6 in HEK293 cells. In contrast to the poor yields obtained for human CALHM1, all three placental paralogs expressed high levels of protein. The elution properties of fusions to green fluorescent protein (GFP) extracted in the detergent glycol-diosgenin (GDN) and analyzed by fluorescent size-exclusion chromatography (FSEC) (*Kawate and Gouaux, 2006*) indicate assemblies of high molecular weight (*Figure 3—figure supplement 1A*). We next proceeded with a scale-up of homomeric CALHM2, 4 and 6 channels in HEK293 cells and purified each protein in the detergent GDN. Consistent with FSEC studies, all purified constructs eluted as large oligomers during gel-filtration chromatography, although at different volumes. The highest elution volume was observed for CALHM6, a similar but slightly lower volume for CALHM2 and the smallest volume for CALHM4, thus hinting towards distinct oligomeric organizations of the three proteins with CALHM4 forming larger complexes than CALHM6 (*Figure 3—figure supplement 1B*). The peak fractions were concentrated, vitrified on cryo-EM grids and used for data collection by cryo-electron microscopy (*Figure 3—figure supplements 2–6*, *Tables 1–2*). As expected from the size-exclusion profiles, all paralogs form large and heterogenic multimeric assemblies containing between 10 and 12 subunits. Whereas in the CALHM6 sample a vast majority of these oligomeric channels do not interact (*Figure 3B*, *Figure 3—figure supplement 4*), we have found almost complete dimerization for CALHM4 (*Figure 3A*, *Figure 3—figure supplements 2–3*) and a significant fraction of particles to dimerize in case of CALHM2 (*Figure 3C*, *Figure 3—figure supplement*

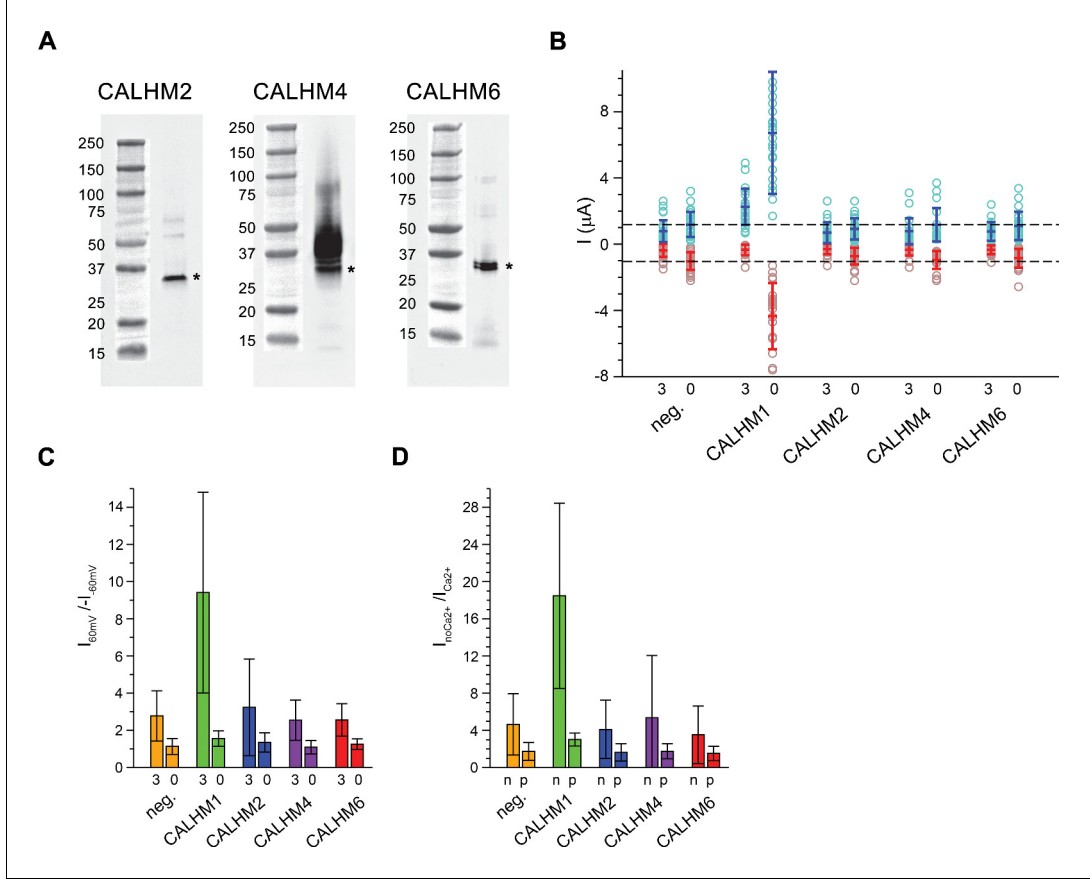

**Figure 2.** Functional characterization of CALHM channels expressed in *X. laevis* oocytes. (**A**) Western blot of proteins located in the plasma membrane of *X. laevis* oocytes heterologously expressing the indicated CALHM channels. Protein was isolated after surface-biotinylation by binding to avidin-resin. The proteins are detected with specific antibodies recognizing the respective CALHM paralog. Left, CALHM2, center CALHM4, right CALHM6. The blot demonstrates the targeting of all three paralogs to the plasma membrane. Bands corresponding to the respective CALHM paralogs are indicated by asterisk. (**B**) Electrophysiological characterization of *X. laevis* oocytes heterologously expressing the paralogs CALHM1, CALHM2, CALHM4 and CALHM6 in comparison to control oocytes (neg.) recorded at extracellular solutions either containing 3 mM $Ca^{2+}$ (3) or 0.5 mM EDTA and 0.5 mM EGTA ($Ca^{2+}$-free, 0). Data show currents of individual oocytes (circle) recorded by two-electrode voltage-clamp (TEVC) at 60 (light blue) and −60 mV (light red). In each case, currents were tabulated at the end of a 5 s voltage step. Averages are shown as bars in red (−60 mV) and blue (60 mV), respectively. Dashed lines indicate mean current levels of control oocytes (neg.) recorded in $Ca^{2+}$-free extracellular solutions at 60 and −60 mV. Currents measured for all investigated channels, except CALHM1, are not significantly different from control oocytes (as judged by a Student t-test). (**C**), Rectification of steady-state currents of oocytes displayed in (**B**) expressed as $I_{60mV}/I_{-60mV}$ calculated for individual oocytes at 3 mM $Ca^{2+}$ (3) and in $Ca^{2+}$-free solutions (0) and averaged. The large value of CALHM1 reflects the activation of the protein at positive voltage in presence of $Ca^{2+}$. (**D**) $Ca^{2+}$-dependence of activation. Change of steady-state currents of oocytes displayed in (**B**) after $Ca^{2+}$-removal expressed as $I_{noCa2+}/I_{Ca2+}$ calculated from individual oocytes at −60 mV (n) and 60 mV (p) and averaged. The large value of CALHM1 at −60 mV reflects the strong activation of currents at negative voltages upon $Ca^{2+}$-depletion. C, D, The difference between the corresponding values of the CALHM paralogs 2, 4, and 6 and neg. are statistically insignificant (as judged by a Student t-test). B-D, Data show averages of 27 (neg.), 21 (CALHM1), 26 (CALHM2), 19 (CALHM4) and 21 (CALHM6) oocytes respectively. Errors are standard deviations.

The online version of this article includes the following figure supplement(s) for figure 2:

**Figure supplement 1.** Electrophysiology traces.

**Table 1.** Cryo-EM data collection, refinement and validation statistics of CALHM4.

| | Dataset 1 CALHM4_Ca$^{2+}$ | | | | Dataset 2 CALHM4_Ca$^{2+}$_free | | |
| --- | --- | --- | --- | --- | --- | --- | --- |
| | 10-mer | | 11-mer | | 10-mer | 11-mer | |
| **Data collection and processing** | | | | | | | |
| Microscope | FEI Tecnai G$^2$ Polara | | | | FEI Tecnai G$^2$ Polara | | |
| Camera | Gatan K2 Summit + GIF | | | | Gatan K2 Summit + GIF | | |
| Magnification | 37,313 | | | | 37,313 | | |
| Voltage (kV) | 300 | | | | 300 | | |
| Electron exposure (e–/Å$^2$) | 40 | | | | 32 | | |
| Defocus range (μm) | -0.8 to -3.0 | | | | -0.8 to -3.0 | | |
| Pixel size (Å) | 1.34 | | | | 1.34 | | |
| Initial particle images (no.) | 422,281 | | | | 576,841 | | |
| Final particle images (no.) | 35,229 | 70,4581* | 27,094 | 54,1881* | 21,264 | 25,703 | 51,4061* |
| Reconstruction strategy2$^†$ | Std | Loc | Std | Loc | Std | Std | Loc |
| Symmetry imposed | D10 | C10 | D11 | C11 | D10 | D11 | C11 |
| Global map resolution (Å) FSC threshold 0.143 | 4.24 | 4.07 | 4.02 | 3.92 | 4.07 | 3.82 | 3.69 |
| Map resolution range (Å) | 4.0-5.1 | 3.8-5.0 | 3.8-5.0 | 3.8-5.0 | 3.8-5.0 | 3.6-4.4 | 3.5-4.3 |
| Map sharpening B factor (Å$^2$) | -200 | -200 | -185 | -177 | -169 | -145 | -126 |
| EMDB identifier | 10920 | 10920*$^‡$ | 10921 | 10921*$^‡$ | 10917 | 10919 | 10919*$^‡$ |
| **Refinement** | | N/A | | N/A | | | N/A |
| Model resolution (Å) FSC threshold 0.5 | 4.2 | | 4.0 | | 4.0 | 3.7 | |
| Model composition Non-hydrogen atoms Protein residues | 41,620 5,340 | | 45,782 5,874 | | 41,620 5,340 | 45,782 5,874 | |
| B factors (Å$^2$) Protein | 62 | | 61 | | 34 | 51 | |
| R.m.s. deviations Bond lengths (Å) Bond angles (°) | 0.005 0.694 | | 0.005 0.672 | | 0.003 0.550 | 0.005 0.754 | |
| Validation MolProbity score Clash score Poor rotamers (%) | 1.53 10.19 0 | | 1.57 6.50 0 | | 1.48 7.59 0 | 1.73 9.79 0 | |
| Ramachandran plot Favored (%) Allowed (%) Disallowed (%) | 98.10 1.90 0 | | 96.66 3.34 0 | | 97.72 2.28 0 | 96.65 3.35 0 | |
| PDB identifier | 6YTO | | 6YTQ | | 6YTK | 6YTL | |

*Subparticles from localized reconstruction.

$^†$Std – standard reconstruction; Loc – localized reconstruction.

$^‡$Higher-resolution map from localized reconstruction submitted as an additional map under the same entry as the main map.

6). Unlike to previous reports of CALHM2 structures, where interactions between channels were mediated by extracellular loops (*Choi et al., 2019*; *Syrjanen et al., 2020*), the pairing in the CALHM4 sample proceeds via contacts at the intracellular side (*Figure 3A*). We also observed two distinct protein conformations in our data with CALHM4 and CALHM6 channels forming cylindrical and conical pores, respectively (*Figure 3D*). Although 2D class averages of CALHM2 appear of high quality, 3D classification of this dataset did not yield high-resolution structures. We believe that the strong preferential orientation of CALHM2 particles resulting in a predominance of views along the

**Table 2.** Cryo-EM data collection, refinement and validation statistics of CALHM6 and CALHM2.

| | Dataset 3 CALHM6_Ca$^{2+}$ | | Dataset 4 CALHM2_Ca$^{2+}$ |
| --- | --- | --- | --- |
| | 10-mer | 11-mer | |
| **Data collection and processing** | | | |
| Microscope | FEI Tecnai G$^2$ Polara | | FEI Tecnai G$^2$ Polara |
| Camera | Gatan K2 Summit + GIF | | Gatan K2 Summit + GIF |
| Magnification | 37,313 | | 37,313 |
| Voltage (kV) | 300 | | 300 |
| Electron exposure (e–/Å$^2$) | 40 | | 55 |
| Defocus range (μm) | -0.8 to -3.0 | | -0.8 to -3.0 |
| Pixel size (Å) | 1.34 | | 1.34 |
| Initial particle images (no.) | 216,859 | | 417,612 |
| Final particle images (no.) | 98,104 | 63,310 | N/A |
| Reconstruction strategy1* | Std | Std | Std |
| Symmetry imposed | C10 | C11 | C1 |
| Global map resolution (Å) FSC threshold 0.143 | 4.39 | 6.23 | N/A |
| Map resolution range (Å) | 4.3-5.1 | 5.0-7.0 | N/A |
| Map sharpening B factor (Å$^2$) | -259 | -435 | N/A |
| EMDB identifier | 10924 | 10925 | N/A |
| **Refinement** | | | N/A |
| Model resolution (Å) FSC threshold 0.5 | 4.4 | 6.6 | |
| Model composition Non-hydrogen atoms Protein residues | 19,560 2,520 | 21,516 2,772 | |
| B factors (Å$^2$) Protein | 85 | 86 | |
| R.m.s. deviations Bond lengths (Å) Bond angles (°) | 0.004 0.773 | 0.004 0.822 | |
| Validation MolProbity score Clash score Poor rotamers (%) | 2.14 16.18 0 | 2.15 17.75 0 | |
| Ramachandran plot Favored (%) Allowed (%) Disallowed (%) | 93.55 6.05 0.40 | 93.95 5.65 0.40 | |
| PDB identifier | 6YTV | 6YTX | |

*Std – standard reconstruction; Loc – localized reconstruction.

pore axis combined with sample heterogeneity has limited our data processing workflow to 2D classification (*Figure 3—figure supplement 6*). In general, we found an oligomeric distribution of CALHM2 channels that corresponds to previously determined structure with a majority of channels being organized as undecamers and a smaller population showing dodecameric assemblies (*Figure 3C*, *Figure 3—figure supplement 6*). Due to the higher quality of the CALHM4 and 6 samples, we continued to use structures derived from these proteins for a detailed characterization of both pore conformations.

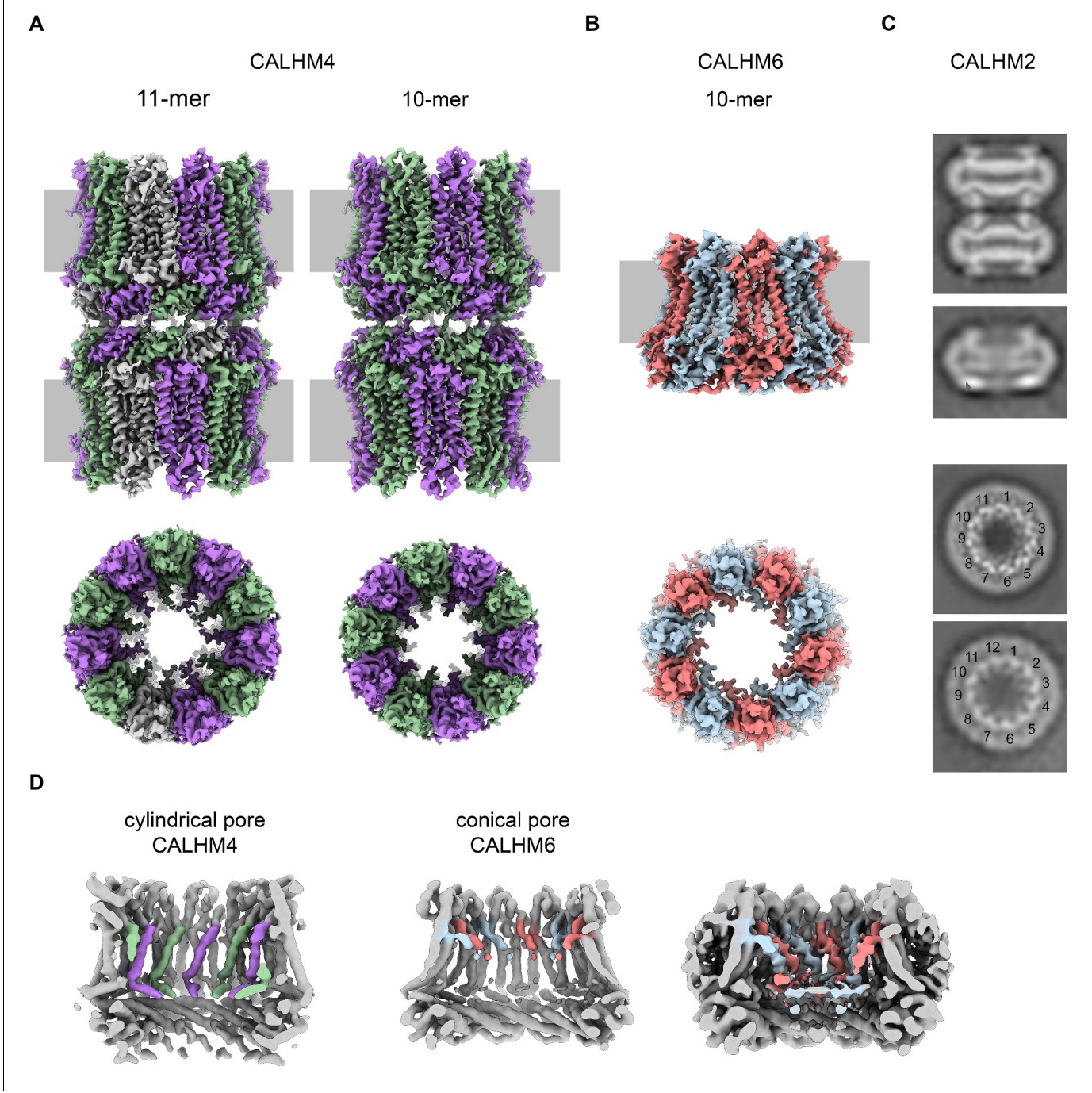

**Figure 3.** Cryo-EM analysis. (**A**) Cryo-EM density of undecameric (11-mer) and decameric (10-mer) pairs of CALHM4 channels at 3.8 and 4.1 Å respectively. Data were recorded from a $Ca^{2+}$-free sample. Subunits are colored in lilac and green, respectively. (**B**) Cryo-EM density of decameric CALHM6 channels at 4.4 Å. Subunits are colored in red and light-blue, respectively. A, B, Views are from within the membrane with membrane indicated as grey rectangle (top) and from the outside (bottom). (**C**) Selected 2D classes of the CALHM2 data showing interacting channel pairs and single channels viewed from within the membrane (top) and views of undecameric and dodecameric channels with subunits numbered (bottom). (**D**) Slices through the CALHM4 (left) and the CALHM6 (right) channels illustrating the distinct features of the cylindrical and conical pore conformations. View of CALHM6 at lower contour (right) shows extended density for the mobile TM1. Maps are low-pass filtered at 6 Å. Colored features refer to density corresponding to TM1 and NH.

The online version of this article includes the following figure supplement(s) for figure 3:

**Figure supplement 1.** Biochemical characterization.

**Figure supplement 2.** Cryo-EM reconstruction of CALHM4 in presence of $Ca^{2+}$.

*Figure 3 continued on next page*

## Cylindrical pore conformation of the CALHM4 structure

For the determination of the CALHM4 structure, we have recorded cryo-EM data in the presence and absence of $Ca^{2+}$ and observed similar structural properties in both samples, which are not affected by divalent cations (*Figure 3—figure supplements 2* and *3*). In each case, we found a heterogeneous, about equal distribution of particles with two distinct oligomeric states. The smaller particles are composed of decameric and the larger of undecameric assemblies, both of which we refer to as CALHM4 channels (*Figure 4*). Due to the slightly higher quality of the $Ca^{2+}$-free sample, we continued to use this data for our further analysis unless specifically indicated. The cryo-EM density of the decamers extends to a resolution of 4.1 Å and undecamers to 3.7 Å, which in both cases permitted the unambiguous interpretation by an atomic model (*Figure 3—figure supplements 2, 3* and *5A*, *Video 1*, *Table 1*). In each structure, the subunits are arranged around a central axis of symmetry that defines the ion-conduction path. As described before, both assemblies contain pairs of CALHM4 channels of the same size related by two-fold symmetry that interact via their cytoplasmic parts (*Figure 4—figure supplement 1A–C*). The CALHM4 channels form approximately 90 Å high cylindrical proteins that span the lipid bilayer with regions at their respective periphery

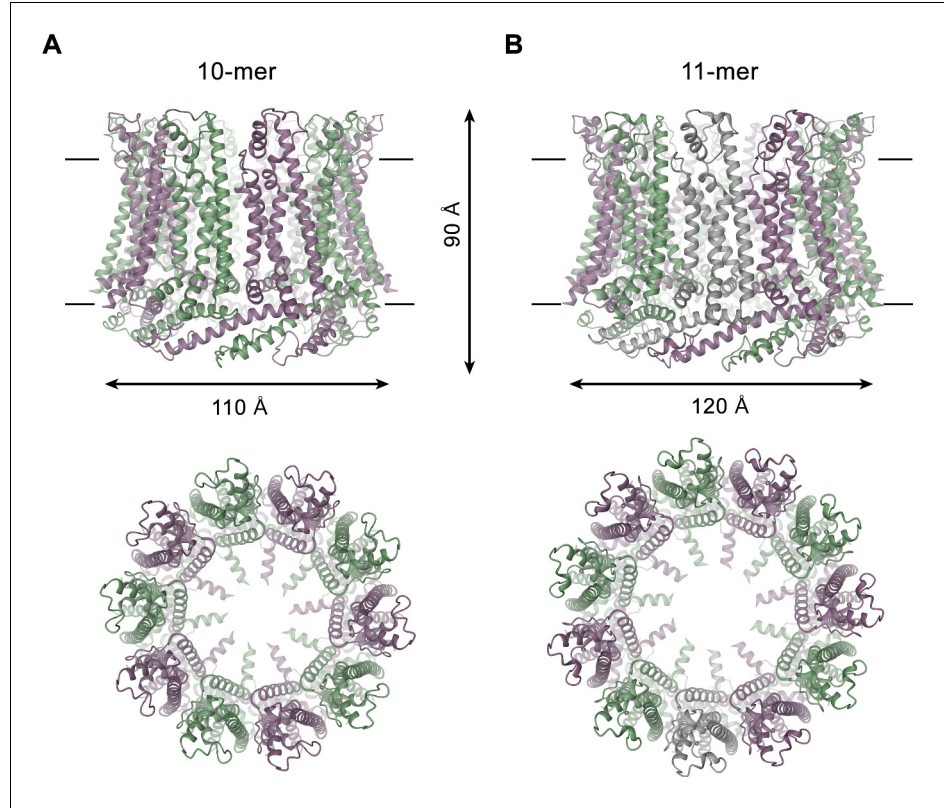

**Figure 4.** CALHM4 structure. Ribbon representation of (**A**), decameric and (**B**), undecameric CALHM4 channels. Top views are from within the membrane with membrane boundaries indicated, bottom views are from the extracellular side. The approximate dimensions are indicated. Subunits are colored in lilac and green.

The online version of this article includes the following figure supplement(s) for figure 4:

**Figure supplement 1.** Features of the CALHM4 structure.

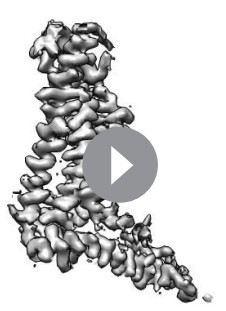

**Video 1.** Cryo-EM density map of a single subunit of CALHM4 obtained in absence of Ca²⁺. Shown is the cryo-EM map of the protein in detergent with the atomic model superimposed.

https://elifesciences.org/articles/55853#video1

extending into the aqueous environment on either side of the membrane (*Figure 4*). In the plane of the membrane, the dimensions of decameric and undecameric CALHM4 channels amount to 110 Å and 120 Å respectively (*Figure 4*, *Figure 4—figure supplement 1D–E*). Irrespective of their distinct oligomerization, the individual subunits in both assemblies show equivalent conformations, which resemble the recently described structures of CALHM1 and 2 (*Choi et al., 2019*; *Demura et al., 2020*; *Syrjanen et al., 2020*) and thus define an organization that is likely general for the CALHM family (*Figure 5A*, *Figure 5—figure supplement 1*). Although the members of this family were predicted to share their architecture with connexins and related innexin and LRRC8 channels (*Siebert et al., 2013*), this turns out not to be

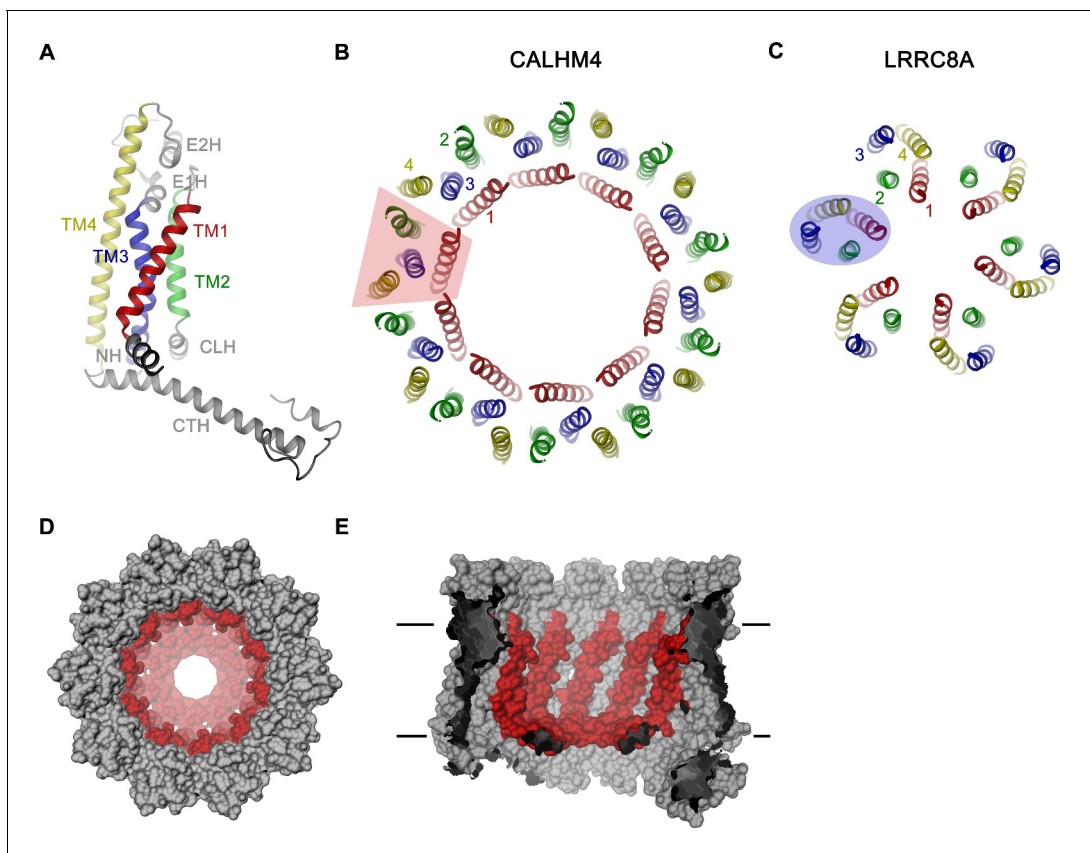

**Figure 5.** CALHM4 subunit and oligomeric arrangement. (**A**) Ribbon representation of the CALHM4 subunit. Secondary structure elements are labeled and transmembrane α-helices are shown in unique colors. View of the transmembrane α-helices of (**B**), the CALHM4 decamer and (**C**), the volume regulated anion channel LRRC8A from the extracellular side. B-C, color code is as in A, transmembrane segments of one subunit are numbered. The general shape of a single subunit is indicated (B, trapezoid, C, oval). (**D**) Surface representation of the CALHM4 decamer. The view is from the outside. (**E**), Slice through the CALHM4 pore viewed from within the membrane. D, E, TM1 and the N-terminal α-helix NH are colored in red.

The online version of this article includes the following figure supplement(s) for figure 5:

**Figure supplement 1.** Sequence and topology.
**Figure supplement 2.** Features of the CALHM4 structure.

the case. Whereas subunits in all families contain four transmembrane helices, which run perpendicular to the membrane, the mutual arrangement of these α-helices differs between CALHM channels and their connexin-like counterparts. When viewed from the extracellular side, the transmembrane helices in connexin, innexin and VRAC channels (*Deneka et al., 2018*; *Kasuya et al., 2018*; *Kefauver et al., 2018*; *Maeda et al., 2009*; *Oshima et al., 2016*) are arranged in a clockwise manner, whereas the arrangement in CALHM channels is anticlockwise (*Figure 5B,C*). Moreover, in connexins and related channels, the four helices form a tightly interacting left-handed bundle with a common core, resulting in a structure with an oval cross-section (*Figure 5C*). Conversely, the helices of the CALHM subunit are organized as two layers conferring an overall trapezoid cross-section (*Figure 5B*). Finally, whereas connexins and LRRC8 channels form hexamers, and innexins form octamers, an oligomeric arrangement that has also been observed for CALHM1, the larger CALHM4 channels contain either 10 or 11 subunits (*Figure 4A,B*). The outer layer of the CALHM subunit is composed of the interacting α-helices TM2-4, which are arranged in one row sharing mutual interactions only with their respective neighbor. The inner layer consists of TM1, which on its entire length exclusively interacts with TM3, the central helix of the outer layer (*Figure 5A,B*). When assembled into oligomers, the helices of the outer layer form a ring, which defines the boundaries of the channel (*Figure 5B*). In this outer ring, the peripheral helices TM2 and 4 are involved in extended interactions with neighboring subunits resulting in a tightly packed interface. Apart from a small fenestration between TM3 and 4 in the center of each subunit, this structural unit shields the interior of the pore from the surrounding membrane (*Figure 5—figure supplement 2A*). In contrast, the respective TM1 helices forming the inner layer are distant from each other and thus not involved in mutual inter-subunit interactions (*Figure 5D,E*). In the region preceding TM1, the residues of the N-terminus form a helix (NH) that is oriented perpendicular to the first transmembrane segment parallel to the plane of the lipid bilayer (*Figure 4*, *Figure 5D*, *Figure 3—figure supplement 5A* and *Figure 5—figure supplement 2B*). Among the family members of known structures, the conformations of NH and TM1 are best defined in CALHM4 (*Figure 3—figure supplement 5A*), whereas weaker density of these fragments in the equivalent state of CALHM1 and 2 points towards a higher mobility of this region in latter proteins (*Choi et al., 2019*; *Syrjanen et al., 2020*). On the extracellular side, a short loop bridges α-helices TM1 and 2 and an extended region containing two short α-helices (E1H and E2H), connects TM3 with the long TM4, which extends beyond the membrane plane (*Figure 5A*). Both segments of the extracellular domain are stabilized by two conserved disulfide bridges (*Figure 5—figure supplement 2C*). On the intracellular side, TM2 precedes a short helix (CLH) that projects away from the pore axis with TM3 being bent in the same direction (*Figure 5—figure supplement 2D*). Both α-helices are connected by a 12-residue long loop that is probably mobile and thus not defined in the density. Downstream of TM4, we find an extended intracellular region. The first halve of this region consists of a long α-helix (CTH), which is tilted by 70° towards the membrane plane. By mutual interaction with neighboring subunits, the CTH-helices form a 30 Å-high intracellular ring that extends from the membrane into the cytoplasm (*Figures 4* and *5A*, *Figure 4—figure supplement 1C*). Distal to CTH, a weakly defined 62-residue long extended loop, which contains interspersed secondary structure elements, folds back towards the intracellular ring and extends to the juxtaposed CALHM4 channel (*Figure 4—figure supplement 1B,C*). In this way the C-terminal loops mediate the bulk of the interaction relating CALHM4 channel pairs in an arrangement whose relevance in a cellular context is still ambiguous.

## Pore architecture

Decameric and undecameric CALHM4 channels contain wide pores, which are cylindrical throughout except for a constriction at the intracellular membrane boundary. The diameter at both entrances of the pore measures about 52 Å and 60 Å for decameric and undecameric assemblies respectively, thus defining the properties of an unusually large channel, which could be permeable to molecular substrates (*Figure 6A*, *Figure 4—figure supplement 1D,E*). Even at the constriction located at the intracellular membrane leaflet where the respective N-terminal helices NH project towards the pore axis, the decameric channels are about 20 Å and undecameric channels 30 Å wide (*Figure 6A*). In both cases the pore would thus be sufficiently large to accommodate an ATP molecule, consistent with the notion of CALHM proteins forming ATP-permeable channels (*Figure 4—figure supplement 1D,E*). Remarkably, this large pore size is in sharp contrast to our functional characterization by electrophysiology where we did not observe appreciable currents for CALHM4 (*Figure 2A*, *Figure 2—*

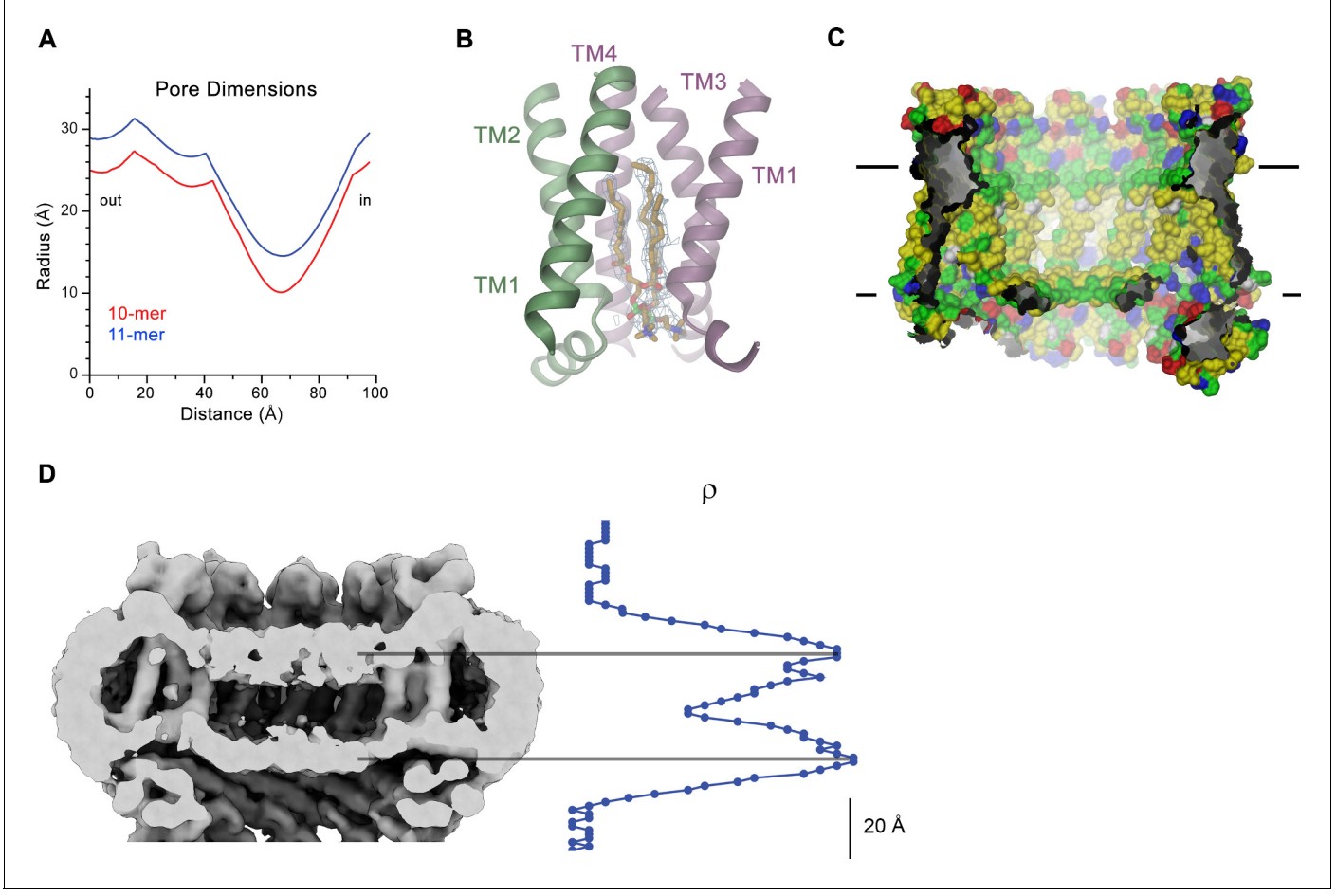

**Figure 6.** CALHM4 pore properties and lipid interactions. (**A**), Pore radius of the CALHM4 decamer (red) and undecamer (blue) as calculated in HOLE (**Smart et al., 1996**). (**B**) Two phosphatidylcholine molecules modeled into residual cryo-EM density (blue mesh) in a cavity at the interface between neighboring α-helices TM1. Secondary structure elements are indicated. (**C**) Chemical properties of residues lining the pore of the CALHM4 channel. Shown is a slice through the pore viewed from within the membrane. The protein is displayed as molecular surface. Hydrophobic residues are colored in yellow, polar residues in green, acidic residues in red and basic residues in blue. (**D**) Slice through the pore region of the CALHM4 undecamer viewed from within the membrane. Shown is non-averaged density in a single copy of the undecameric CALHM4 channel pair at low contour to highlight the location of increased density within the pore corresponding to a bilayer of either phospholipids or detergents. A plot of the density along the pore axis showing two maxima that are separated by the expected distance between the headgroup regions of a lipid bilayer is shown right. B, D, Displayed cryo-EM density refers to data from the undecameric channel in presence of $Ca^{2+}$.

The online version of this article includes the following figure supplement(s) for figure 6:

**Figure supplement 1.** Lipid interactions in the pore of CALHM4.

**Figure supplement 2.** LC-MS analysis of co-purified lipids.

figure supplement 1E), which suggests that either the absence of activating or presence of inhibiting components might impede ion conduction.

Within the membrane, the pore diameter of CALHM4 is confined by the ring of non-interacting TM1 helices, which is placed inside an outer ring of the channel formed by helices TM2-4 (**Figure 5B,D**). This arrangement creates clefts between neighboring helices of the inner ring which are delimited by helices TM2 and 4 at the respective subunit interfaces (**Figure 5B,E**). In our sample, this cleft appears to be filled with lipids as indicated by the residual density observed in the cryo-EM maps (**Figure 6B**, **Figure 6—figure supplement 1A,B**). Due to the prevalence of aliphatic residues lining the pore at the assumed location of the membrane core, the pore is highly hydrophobic, whereas the regions extending into the aqueous environment contain polar and charged residues (**Figure 6C**). Similarly, the side of the helical N-terminus, which faces the membrane at its intracellular boundary, is hydrophobic (**Figure 5—figure supplement 2B**). In light of its large cross-section

and high hydrophobicity, it is conceivable that the interior of the pore would accommodate lipids, which potentially could form bilayers that would restrict ion permeation as recently suggested for CALHM2 based on molecular dynamics simulations (*Syrjanen et al., 2020*). In our data, we find strong evidence for a layered distribution of density inside the pore within the presumed membrane region in non-symmetrized maps and after application of symmetry. This density is observed with similar properties in decameric and undecameric proteins in both datasets of CALHM4 obtained in either absence or presence of Ca$^{2+}$ (*Figure 6D*, *Figure 6—figure supplement 1C*). Its distribution displays features that quantitatively match the corresponding properties of lipid membranes obtained from a comparison to cryo-EM density of liposomes and computer simulations (*Figure 6—figure supplement 1D,E*) and could thus either reflect the presence of lipids or detergent molecules arranging in a bilayer-like structure facilitated by the confined pore geometry. We thus analyzed the composition of small molecules that are co-purified with CALHM4 by mass spectrometry and were able to detect phospholipids that are commonly found in the membranes of HEK cells (*Figure 6—figure supplement 2*). Collectively, our data are compatible with the presence of a lipid bilayer located within the pore region of CALHM4 channels, which could interfere with the diffusion of charged substances. Thus, despite of its large pore diameter, it is at this point unclear whether the CALHM4 structure defines a conductive conformation of CALHM channel or alternatively a conformation that harbors a membrane-like assembly residing inside the pore that would impede ion conduction.

## Conical pore conformation of the CALHM6 structure

As in case of CALHM4, data of CALHM6 shows a heterogeneous distribution of decameric and undecameric channels, but in this case with a prevalence of the former amounting to 60% of the classified particles (*Figure 3—figure supplement 4*). Both assemblies of CALHM6 contain subunits with equivalent conformations, which are better defined in the smaller oligomers (*Figure 3B*, *Figure 3—figure supplement 4*). We have thus chosen the decameric CALHM6 structure for a description of the conical pore of a CALHM channel (*Figures 3D* and *7A–C*). Unlike CALHM4, the lower resolution of the CALHM6 density of 4.4 Å, precludes a detailed interpretation of the structure for all parts of the protein. Still, the high homology between the two paralogs and density attributable to large side chains constrains the placement of helices and conserved loop regions and thus allows the credible analysis of major conformational differences (*Figure 3—figure supplement 5B*, *Video 2*, *Table 2*). Unlike CALHM4, there is no dimerization of CALHM6 channels and the C-terminal region following the cytoplasmic helix CTH instead appears to engage in intramolecular interactions with the outside of the cytoplasmic rim for most of its length (*Figure 7A*). This observation further supports the notion that the dimerization of CALHM4 might be a consequence of interactions formed between solubilized proteins where the mobile C-terminus could equally well engage in intra- and intermolecular interactions, of which the latter would be multiplied in the highly symmetric arrangement. Although the general organization of the CALHM6 channel closely resembles CALHM4, it shows a distinctly different state of the pore (*Figure 3D*). Upon comparison of decameric channels of both paralogs, in CALHM6 we find a slight expansion of the protein parallel to the membrane and an accompanying moderate contraction in perpendicular direction (*Figure 7—figure supplement 1A*, *Video 3*). In a superposition of the subunits, similar conformations are observed for α-helices TM2 and 4 and the C-terminal helix CTH and larger differences in the interacting helices TM1 and 3 (*Figure 7D*, *Figure 7—figure supplement 1B,C*). These differences are most pronounced for TM1, which in CALHM6 has detached from TM3 and moved by 60° towards the pore axis around a hinge located upstream of a conserved phenylalanine at the end of TM1 (*Figure 7B–E*). In both structures, the conformation of the proximal loop connecting TM1 and 2 is stabilized by two conserved cysteines, which are involved in disulfide bridges with the region connecting TM3 and TM4 (*Figure 7F–G*, *Figure 5—figure supplement 2C*, *Video 4*). As a consequence of the disruption of its interaction with TM1, the intracellular halve of TM3 tilts away from the pore axis by 30° around a pivot located close to a conserved proline residue (P115 in CALHM4), which probably destabilizes the helix (*Figure 7—figure supplement 1B,D*). The transition from a conformation observed in CALHM4 to a conformation defined by CALHM6 is accompanied by the dissociation of interactions between TM1 and 3, which are mediated by conserved residues involving a cluster of hydrophobic interactions at the extracellular side and additional interactions on the entire length of both α-helices (*Figure 7F–G*, *Figure 7—figure supplement 1C,D*, *Video 4*). The iris-like movement of TM1 in the

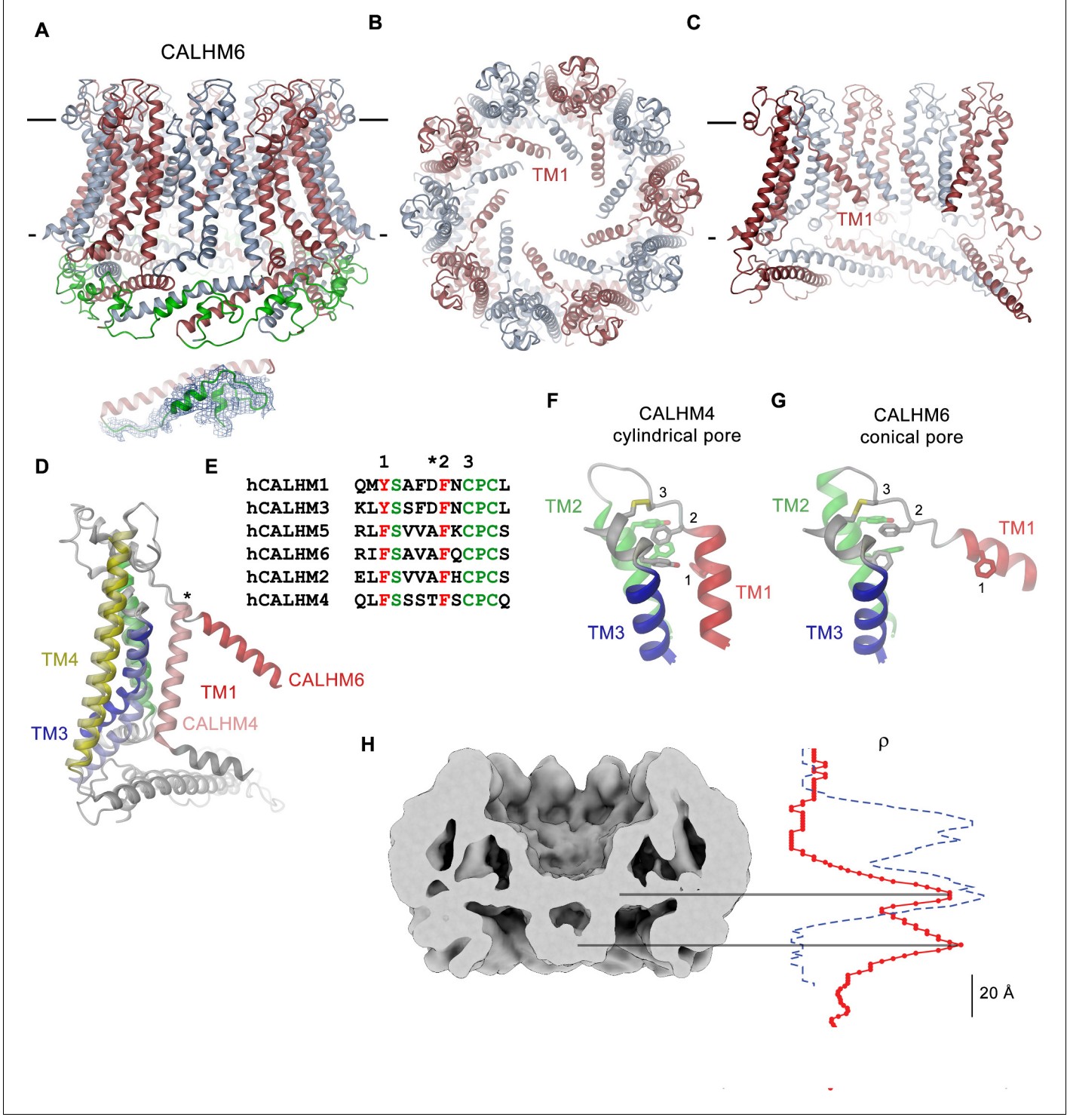

**Figure 7.** CALHM6 structure. (**A**) Ribbon representation of the decameric CALHM6 structure viewed from within the membrane. Subunits are colored in red and light blue. The C-terminus following CTH is colored in green. Inset (below) shows a close-up of this region with cryo-EM density superimposed. (**B**) View of the CALHM structure from the extracellular side. (**C**) Slice through the pore of the CALHM6 structure. B, C, TM1, which has moved towards the pore axis is labeled. A, C, the membrane boundary is indicated. (**D**) Superposition of single subunits of the CALHM4 and CALHM6 structures illustrating conformational changes. Coloring is as in *Figure 5A* with CALHM4 shown in brighter shades of the same color. Secondary structure elements are labeled and the hinge for the movement of TM1 is indicated by an asterisk. (**E**) Sequence alignment of the end of TM1 and the following loop of CALHM paralogs. Selected conserved residues are colored in green and red. Numbering corresponds to residues indicated in panels F and G. Asterisk marks the hinge region displayed in D. Close-up of the extracellular region involved in conformational changes in (**F**), the cylindrical

*Figure 7 continued on next page*

*Figure 7 continued*

conformation displayed in CALHM4 and (**G**), the conical conformation displayed in CALHM6. Residues contributing to a cluster of aromatic residues on TM1-3 and a conserved disulfide bridge are shown as sticks. Selected positions highlighted in E are labeled. F, G, Coloring is as in *Figure 5A*. (**H**) Slice through the pore region of the CALHM6 decamer viewed from within the membrane. Shown is non-averaged density at low contour to highlight the location of diffuse density within the pore. A plot of the density along the pore axis of CALHM6 is shown in red, the corresponding density in CALHM4 is shown as a dashed blue line for comparison. The two maxima in the CALHM6 density are shifted towards the intracellular side. The density corresponding to the headgroups of the outer leaflet of the bilayer in CALHM4 is absent. Density at the location of the headgroup region at the inner leaflet of the bilayer and further towards the intracellular side could correspond to either lipids or to the poorly ordered N-terminus.

The online version of this article includes the following figure supplement(s) for figure 7:

**Figure supplement 1.** Features of the CALHM6 structure.

oligomeric channel in its transition between conformations probably requires concerted rearrangements to avoid steric clashes in the crowded environment of the pore. When viewed from the outside this transition thus resembles the closing of an aperture and it converts a cylindrical pore to a funnel which narrows towards the intracellular side while creating a large cavity between TM1 and TM3 that becomes accessible from the cytoplasm (*Figure 7C*, *Video 3*). In the cryo-EM density, TM1 is less well defined compared to the rest of the protein reflecting its increased flexibility in the observed structure (*Figure 3—figure supplement 5B*). Since the conformation of NH, which has moved towards the pore axis, is not defined in the density, the size of the CALHM6 channel at its constriction remains ambiguous. It could range from an occluded pore if NH helices make contacts in the center of the channel (modeled as clogged conformation) to a pore with similar diameter as found at the CALHM4 constriction. The latter could be obtained in case the mutual relationship between TM1 and NH remains unchanged compared to the CALHM4 structure (defined as kinked conformation) or if NH straightens in continuation of TM1 (in an extended conformation) (*Figure 7—figure supplement 1E*). In any case, the large conformational changes would affect the location of lipids within the pore, which in case of an internal bilayer would have to rearrange in response to the severely altered pore geometry. Such rearrangement is reflected in the changed distribution of the residual electron density inside the pore (*Figure 7H*). Whereas, compared to CALHM4, we find density at the location of the intracellular layer of lipids and further towards the cytoplasm, part of which might be attributable to the mobile N-terminal α-helix NH, the outer layer of density corresponding to the putative extracellular leaflet of a bilayer has disappeared. A comparable distribution of pore density is found in the structure of CALHM2 in complex with ruthenium red, which resides in a similar conical pore conformation (*Choi et al., 2019*). Thus, despite the pronounced conformational differences to CALHM4 and the fact that the CALHM6 structure appears to contain features of a closed pore, a definitive functional assignment remains also in this case ambiguous.

## Discussion

In the presented study, we have addressed the structural and functional properties of CALHM channels in the human placenta. To identify relevant CALHM paralogs in this organ, we have quantified their expression in samples of the whole placenta and in isolated trophoblast cells and found CALHM2, 4 and 6 to be abundant on a transcript level (*Figure 1*). Their high expression compared to other membrane proteins (*Figure 1—figure supplement 1*) suggests an important functional relevance of these membrane channels in mediating transport processes during the development of the fetus, although their detailed role still awaits to be explored.

The structural characterization of the three placental paralogs by cryo-electron microscopy has provided insight into fundamental features

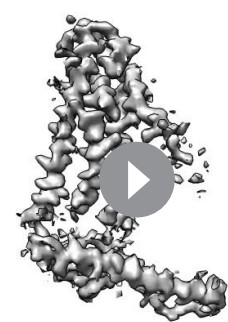

**Video 2.** Cryo-EM density map of a single subunit of CALHM6. Shown is the cryo-EM map of the protein in detergent with the atomic model superimposed.
https://elifesciences.org/articles/55853#video2

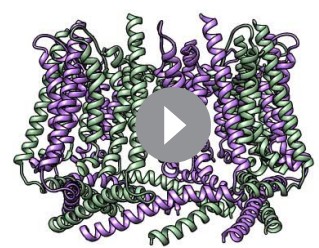

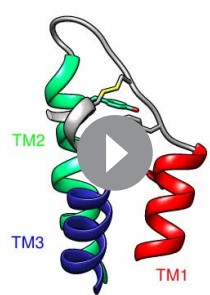

**Video 3.** Morph between the cylindrical pore conformation of CALHM4 and the conical pore conformation displayed in the 'extended' model of CALHM6. In the latter the N-terminal α-helix NH is modeled in continuation of TM1 resulting in a constricting pore diameter similar to CALHM4.
https://elifesciences.org/articles/55853#video3

**Video 4.** Morph of the extracellular region of a single CALHM subunit involved in conformational changes between the cylindrical conformation displayed in CALHM4 and the conical conformation displayed in CALHM6. Residues contributing to a cluster of aromatic residues on TM1-2 and a conserved disulfide bridge are shown as sticks. Coloring is as in *Figure 5A*.
https://elifesciences.org/articles/55853#video4

of the family which largely conform with properties that have recently been described for CALHM1 and 2 (*Choi et al., 2019*; *Demura et al., 2020*; *Syrjanen et al., 2020*) but which also show substantial differences. Similar to previous studies, our structures have defined the architecture of the CALHM subunit which, although containing the same number of membrane-spanning helices as VRACs and gap-junctions forming connexins, innexins and pannexins, is distinct from these proteins (*Figure 5B,C*). The CALHM subunits thus constitute unique modular building blocks that assemble with different stoichiometries into large membrane channels. In contrast to previously described CALHM structures, where the oligomeric state of individual channels was described as uniform, our data show heterogeneous populations with CALHM4 and CALHM6 assembling as decameric and undecameric channels with similar abundance and CALHM2 as predominantly undecameric proteins with a smaller population of dodecamers (*Figure 3*, *Figure 3—figure supplements 2–4* and *6*). The diverse oligomerization of different paralogs is generally consistent with the assemblies observed for the previously described CALHM1 and CALHM2 structures, which were reported to form octamers and undecamers, respectively (*Choi et al., 2019*; *Demura et al., 2020*; *Syrjanen et al., 2020*), and it underlines the ability of CALHM proteins to constitute membrane channels of different sizes. Although the physiological role of these different oligomeric states is currently unclear, the observed heterogeneity might reflect the low energetic penalty for the incorporation of additional subunits into large oligomeric channels of the CALHM family. In this respect, the smaller size of CALHM1 channels could be responsible for its functional properties, which are manifested in electrophysiological recordings. Whereas CALHM1 in our hands showed the previously described functional hallmarks of a channel that is activated by depolarization and removal of extracellular calcium, we have not observed pronounced activity of CALHM2, 4 and 6 under the same conditions, despite their efficient targeting to the plasma membrane (*Figure 2*, *Figure 2—figure supplement 1*). The low current response contrasts with the larger oligomeric organization of these proteins, which should lead to channels of even higher conductance than observed for CALHM1, and thus likely reflects their low open probability. Together our findings suggest that CALHM2, 4, and 6 are regulated by distinct, still unknown mechanisms. In that respect it remains puzzling how the large pores observed for the investigated structures (with diameters of CALHM4 decamers exceeding 40 Å within the membrane and 20 Å at the respective constriction, *Figure 6A*) can be regulated to prevent leakage of substances under resting conditions, which would be deleterious to the cell. Despite the unknown activating stimuli, our study has provided insight into gating transitions of CALHM channels by showing two conformations with either a cylindrical pore of uniform large dimeter within the membrane that is constricted at the intracellular side as in case of CALHM4 and a conical pore that continuously

narrows from its extracellular entry towards the cytoplasm as in case of CALHM6 (*Figures 4* and *7*). The main difference in the pore geometry results from a large rearrangement of the pore-lining α-helix TM1 which is accompanied by a smaller change in TM3 (*Figure 7D*). Whereas in the cylindrical pore conformation of CALHM4, TM1 tightly interacts with TM3, which is a part of a densely packed outer rim of the pore consisting of TM2-4, the helix has dissociated from its interaction site and instead moved towards the symmetry axis to alter the pore geometry in the conical conformation of CALHM6. Similar conformational properties have previously been described for CALHM2 channels, although with a different orientation of TM1 in the conical pore conformation (*Choi et al., 2019*; *Figure 8—figure supplement 1A–C*). In this previous study, the cylindrical conformation was assigned to an open state and the conical conformation to a closed state of the pore, a proposal that is further supported by a structure of the bound blocker rubidium red which appears to stabilize the conical conformation (*Choi et al., 2019*; *Figure 8*). While, at first glance, the relationship between observed pore conformations and their corresponding functional states appears evident, there are still puzzling questions which prevent a definitive assignment at this stage. For example in light of the large diameter of the CALHM4 pore, the poor conductance properties in functional recordings remain mysterious. In that respect, the presence of lipids within the pore of large CALHM channels is noteworthy as it offers a potential alternative mechanism for regulation. In our structure of CALHM4, we find bound lipids inside the pore to stabilize the gap between non-interacting TM1 helices from different subunits (*Figure 6B*). Additionally, the bimodal diffuse residual cryo-EM density within the hydrophobic interior of CALHM4, which either originates form detergents or co-purified lipids that assemble as bilayers in the constrained environment of the pore are remarkable (*Figure 6D*). This distribution of density has changed markedly in the conical structure of CALHM6 (*Figure 7H*) and hints at a potential role of lipids in shaping the activation and permeation properties

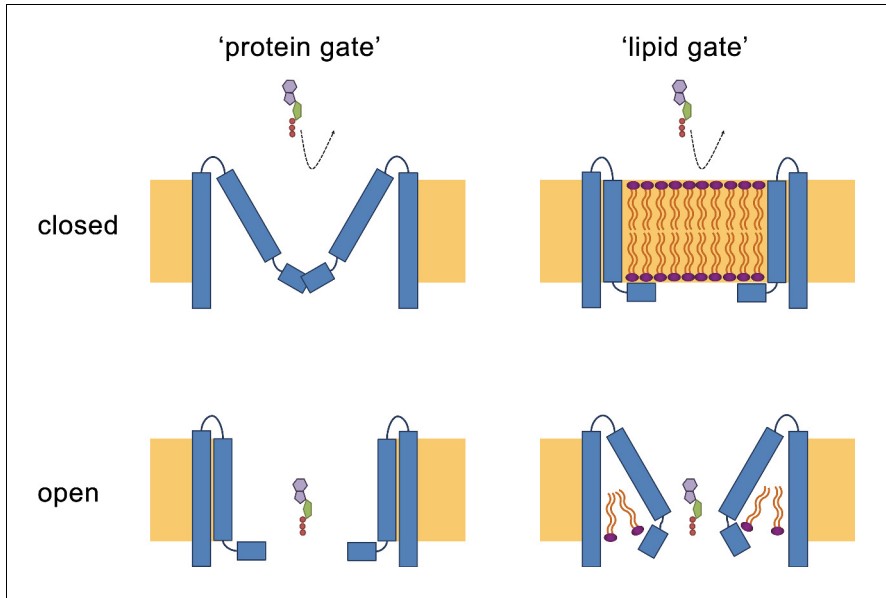

**Figure 8.** Hypothetical gating mechanisms. Schematic illustration of hypothetical gating mechanisms of large CALHM channels. Left, 'protein gate'. The gate impeding ion conduction of large CALHM channels (i.e. CALHM2, 4, 6) in the closed state is formed by the N-terminus of the protein, which closes the pore as observed in the modeled 'clogged' conformation of CALHM6 (top). In this case the conical conformation of CALHM6 would display a closed pore and the cylindrical conformation (bottom) an open pore. Right, 'lipid gate'. The gate impeding ion conduction of large CALHM channels in the closed state is formed by lipids assembling as a bilayer within the protein, which impedes ion conduction. Since bilayer formation is facilitated in the cylindrical conformation of CALHM4, this structure represents a closed pore (top) whereas the structure of the bilayer would be disturbed in the conical conformation of CALHM6 (bottom). Since both the 'kinked' and 'extended' pore conformations of CALHM6 show a large opening, these conformations could represent open pores.

The online version of this article includes the following figure supplement(s) for figure 8:

**Figure supplement 1.** Comparison of pore conformations.

of large CALHM channels, which is currently still not understood (*Figure 8*). We thus could envision two alternative scenarios which might underly regulation in large CALHM channels where the gate in the closed state could either be formed by part of the protein or by lipids (*Figure 8*, *Figure 8—figure supplement 1D*). Both models are at this stage hypothetical and it is still unclear how lipids in the pore would rearrange during activation. Another remarkable feature of large CALHM channels concerns their tendency to form pairwise assemblies upon extraction from the membranes (*Figure 3A,C*, *Figure 3—figure supplements 2*, *3* and *6*). This property has previously been observed for undecameric CALHM2 channels, which were described to dimerize on the extracellular side thus forming structures that resemble gap-junctions, and in the present study, where dimerization was found to some degree in all samples. In our data, the described behavior is most pronounced in case of CALHM4 where dimerization proceeds on the intracellular side mediated by interactions with the partly mobile C-terminus. Although intriguing, it is currently unclear whether any of the observed pairwise interactions is of relevance in a physiological context.

Thus, although our data have provided a large step forward towards the comprehension of CALHM channels, it has also opened many questions. An important area of future investigations relates to the characterization of the localization of CALHM channels in different placental cell types and the identification of their subcellular distribution. Knowledge of their localization would provide a first glimpse into potential roles of these large channels for transport processes in this organ and should provide answers on the potential relevance of intra- and extracellular interactions of channel pairs. Another question concerns the ability of the three paralogs to heteromerize and the potential relevance of such heteromeric channels in a physiological environment. Finally, it will be important to study the activation mechanism of the described channels and whether they are embedded in larger interaction networks, which shape their activation properties. These topics will be addressed in future studies for which our current data provide an important foundation.

# Materials and methods

## Placenta collection
Human placental tissues were collected from the Division of Obstetrics and Gynecology, Lindenhofgruppe Bern, Switzerland, under approval by the ethical commission of the Canton of Bern (approval No. Basec 2016–00250). Written informed consent was obtained from all participants. Placentas were collected from uncomplicated pregnancies following elective cesarean section beyond 37 weeks of gestation without prior labor upon patients request or due to breech presentation. All experiments were carried out in accordance with the relevant guidelines and regulations.

## Expression analysis in placental tissue
### RNA isolation, reverse transcription and quantitative RT-PCR
Approximately 50 mg of frozen placental tissue was subjected to RNA isolation as previously described (*Huang et al., 2018*; *Huang et al., 2013*). All RNA samples included in the study had an OD260/280 ratio >1.8. First-strand cDNA was synthesized from 2 µg of total RNA with oligo $(dT)_{15}$ primers and GoScript Reverse Transcriptase (Promega, Switzerland) according to the manufacturer's instructions. The qPCR reaction in SYBR Green reagent (10 µl) contained 0.5 µM primers, 2 x GoTaqqPCR Master Mix (Promega, Switzerland) and 1 µl cDNA. Primer nucleotide sequences and PCR efficiencies are shown in *Figure 1—figure supplement 1C and D*. Amplification reactions were performed in duplicates in 384-well plates with the ViiA7 system (Applied Biosystems, USA). To evaluate mRNA quantities, data were obtained as $C_t$ values (describing the cycle number at which logarithmic plots cross calculated threshold lines). $C_t$ values were used to determine $\Delta C_t$ values ($\Delta C_t = C_t$ value of the reference gene minus the $C_t$ value of the target gene). The applied reference gene was Tyrosine 3-monooxygenase/tryptophan 5-monooxygenase activation protein, zeta polypeptide (YWHAZ). Comparative transcript data between undifferentiated (CTB) and differentiated (STB) cells were calculated as $2^{-\Delta\Delta Ct}$ values ($\Delta\Delta C_t = C_t$ value of CALHM gene – $C_t$ value of YWHAZ) and are presented as x-fold difference. Data analysis and statistical evaluations were performed using paired 2-way ANOVA with Sidak's multiple comparisons test with GraphPad Prism software.

## Primary trophoblast isolation and characterization

Villous trophoblast cells were isolated from healthy human term placentas by enzymatic digestion and gravitational separation as previously described (*Nikitina et al., 2011*; *Huang et al., 2013*) with minor modifications. Briefly, villi-rich tissues were digested three times with 0.25% trypsin (Sigma, USA) and 300 IU/ml Deoxyribonuclease I (Sigma, USA), and then subjected to Percoll (Sigma, USA) density gradient centrifugation to isolate villous cytotrophoblast cells. In order to assure the purity of the cells, flow cytometry analysis was performed by using the trophoblast-specific epithelial cell marker cytokeratin 7 (*Maldonado-Estrada et al., 2004*) (anti-cytokeratin 7, Dako, Switzerland). Vimentin (anti-vimentin, Sigma, USA), which is only expressed in potentially contaminating cells (e.g. mesenchymal cells, fibroblasts, smooth muscle cells, stromal cells) served as a negative marker (*Soares and Hunt, 2006*).

## Primary trophoblast cell culture

Isolated human trophoblast cells were cultured in Dulbecco's modified Eagle's medium containing 4.5 g/l glucose (DMEM-HG, Gibco, UK) supplemented with 10% fetal bovine serum (FBS, Seraglob, Switzerland) and Antibiotic-Antimycotic (Gibco, USA) in a humidified incubator under a 5% $CO_2$ atmosphere at 37°C. Cells were seeded at a density of $0.2 \cdot 10^6$ cells/cm$^2$ in costar CellBIND six well plates (Corning, USA) and harvested after 24 hr (cytotrophoblast stage) or 48 hr (STB stage). The syncytialization process was confirmed by visualization under the microscope.

## Expression cell lines

Adherent HEK293T cells used for protein expression and screening experiments were cultured in DMEM medium supplemented with 10% FBS and penicillin/streptomycin in a humidified incubator under a 5% $CO_2$ atmosphere at 37°C. Suspension-adapted HEK293S GnTI$^-$ cells used for protein expression and purification were grown in HyClone TransFx-H media, supplemented with 2% fetal bovine serum (FBS), 4 mM L-glutamine, 1.5% Poloxamer 188 and penicillin/streptomycin in a humidified incubator under 5% $CO_2$ atmosphere at 37°C. Both cell lines were tested negative for mycoplasma contamination.

For construct generation, cDNAs of human CALHM proteins, with SapI restriction sites removed, were obtained from GenScript. For expression in *X. laevis* oocytes, cDNAs of CALHM1, CALHM2, CALHM4 and CALHM6 were cloned into a pTLNX vector. For expression in mammalian cells, cDNAs of all human CALHM homologs were cloned into a pcDNA 3.1 vector which was modified to be compatible with FX cloning technology (*Geertsma and Dutzler, 2011*) and to encode a C-terminal 3C protease cleavage site followed by Venus-Myc-SBP tag.

## Homolog screening

For overexpression studies of CALHM paralogs, adherent HEK293T cells were seeded on 10 cm dishes and grown to a density of $7 \cdot 10^6$ cells/well. Subsequently, cells were transiently transfected with mixtures of respective *CALHM* cDNA constructs and polyethylenimine with an average molecular weight of 25 kDa (PEI 25K, branched). As a preparative step, 12 µg of DNA and 48 µg of PEI were separately incubated in 0.5 ml of DMEM medium. After 5 min, both solutions were mixed, incubated for 15 min and added to respective cell cultures. 36 hr post transfection cells were washed with PBS, harvested by centrifugation at 400 g for 5 min, flash-frozen in liquid nitrogen and stored at −20°C. For protein extraction, cells were thawed on ice, lysed with lysis buffer (25 mM HEPES, 150 mM NaCl, 0.5 mM CaCl$_2$, 2 mM MgCl$_2$, 2% GDN, protease inhibitors, RNase, DNase, pH 7.6) for 1 hr and clarified by centrifugation at 16,000 g for 10 min. The obtained supernatants were filtered through 0.22 µm centrifugal filters and injected onto a Superose 6 5/150 column equilibrated with elution buffer (10 mM HEPES, 150 mM NaCl, 50 µM GDN pH 7.6) and eluted in the same buffer. Proteins were identified by recording of the fluorescence of the attached Venus YFP.

## Protein expression and purification

Suspension HEK293S GnTI$^-$ cells were grown in Bioreactor 600 vessels (TPP) and seeded to a density of $0.6 \cdot 10^6$ cells/ml a day prior to transfection. For protein expression, cells were transiently transfected with mixtures of *CALHM* cDNA constructs and PEI MAX 40K. For 300 ml of cell culture, 0.5 mg of DNA and 1.2 mg of PEI MAX were suspended in 20 ml volume of DMEM medium. After 5

min the DNA solutions were mixed with the PEI MAX solutions, incubated for 15 min and supplemented with 4 mM valproic acid. Transfection mixtures were subsequently added to cell cultures and expression proceeded for 36 hr. Afterwards, cells were harvested by centrifugation at 500 g for 15 min, washed with PBS and stored at −20°C for further use. All following purification steps were carried out at 4°C. For protein extraction, cells were suspended in lysis buffer (25 mM HEPES, 150 mM NaCl, 1% Lauryl Maltose Neopentyl Glycol (LMNG), 0.5 mM $CaCl_2$, 2 mM $MgCl_2$, protease inhibitors, RNase, DNase, pH 7.6) and incubated for 1 hr under constant stirring. For CALHM4 purification in $Ca^{2+}$-free conditions, $Ca^{2+}$ was replaced by 5 mM EGTA after extraction. To clarify the extracts, lysates were centrifuged at 16,000 g for 15 min. Supernatants filtered through a 0.5 µm filter were applied to the StrepTactin Superflow affinity resin and incubated with the slurry for 2.5 hr under gentle agitation. Unbound proteins were removed and bound proteins were eluted with elution buffer (10 mM HEPES, 150 mM NaCl, 50 µM GDN, 2 mM $CaCl_2$ and 10 mM d-Desthiobiotin, pH 7.6). For purification in $Ca^{2+}$-free conditions, $Ca^{2+}$ was replaced by 2 mM EGTA. For cleavage of fusion tags, eluates were incubated for 30 min with 3C protease. Subsequently, samples were concentrated, filtered through 0.22 µm filters and subjected to size exclusion chromatography on a Superose 6 10/300 GL column equilibrated with SEC buffer (10 mM HEPES, 150 mM NaCl, 50 µM GDN, 2 mM $CaCl_2$/2 mM EGTA, pH 7.6). Peak fractions were pooled and concentrated with 100 kDa MWCO centrifugal filters.

## LC-MS analysis of co-purified lipids

For the analysis of lipids co-purified with CALHM4, the protein was prepared in the presence of $Ca^{2+}$ in the same manner as described for structure determination. For lipid extraction, chloroform was added in a 1:1 ratio (v/v) to 200 µl of a CALHM4 sample with a protein concentration of 3.75 mg ml$^{-1}$ or to 200 µl of SEC buffer as a control (blank). The samples were briefly vortexed and incubated for 5 min at ambient temperature until phases have separated. The lower organic phase was collected and used for liquid chromatography and mass spectrometry (LC-MS) analysis. Prior to analysis, 25 µl of chloroform extract was mixed with 475 µl of a 50% aqueous methanol solution. The suspension was vortexed and centrifuged at 20°C for 10 min at 16,000 g. 100 µl of the supernatant were transferred to a glass vial with narrowed bottom (Total Recovery Vials, Waters) and subjected to LC-MS. Lipids were separated on a nanoAcquity UPLC (Waters) equipped with a HSS T3 capillary column (150 µm x 30 mm, 1.8 µm particle size, Waters), applying a gradient of 5 mM ammonium acetate in water/acetonitrile 95:5 (A) and 5 mM ammonium acetate in isopropanol/acetonitrile 90:10 (B) from 5% B to 100% B over 10 min. The following 5 min conditions were kept at 100% B, followed by 5 min re-equilibration to 5% B. The injection volume was 1 µl. The flow rate was constant at 2.5 µl/min. The UPLC was coupled to a QExactive mass spectrometer (Thermo) by a nano-ESI source. MS data were acquired using positive polarization and data-dependent acquisition (DDA). Full scan MS spectra were acquired in profile mode from 80 to 1,200 m/z with an automatic gain control target of 1 10$^6$, an Orbitrap resolution of 70,000, and a maximum injection time of 200 ms. The five most intense charged (z = +1 or +2) precursor ions from each full scan were selected for collision induced dissociation fragmentation. Precursor was accumulated with an isolation window of 0.4 Da, an automatic gain control value of 5 10$^4$, a resolution of 17,500, a maximum injection time of 50 ms and fragmented with a normalized collision energy of 20 and 30 (arbitrary unit). Generated fragment ions were scanned in the linear trap. Minimal signal intensity for MS2 selection was set to 500. Lipid datasets were evaluated with Progenesis QI software (Nonlinear Dynamics), which aligns the ion intensity maps based on a reference data set, followed by a peak picking on an aggregated ion intensity map. Detected ions were identified based on accurate mass, detected adduct patterns and isotope patterns by comparing with entries in the LipidMaps Data Base (LM). A mass accuracy tolerance of 5 mDa was set for the searches. Fragmentation patterns were considered for the identifications of metabolites. Matches were ranked based on mass error (observed mass – exact mass), isotope similarity (observed versus theoretical) and relative differences between sample and blank.

## Cryo-EM sample preparation and data collection

For structure determination of human CALHM2, 4 and 6 in the presence of $Ca^{2+}$ and of human CALHM4 in the absence of $Ca^{2+}$ by cryo-EM, 2.5 µl samples of GDN-purified proteins at a concentration of 1.5–3 mg ml$^{-1}$ were applied to glow-discharged holey carbon grids (Quantifoil R1.2/1.3 or

R0.6/1 Au 200 mesh). Excess liquid was removed in a controlled environment (4°C and 100% relative humidity) by blotting grids for 4–6 s. Grids were subsequently flash frozen in a liquid propane-ethane mix using a Vitrobot Mark IV (Thermo Fisher Scientific). All samples were imaged in a 300 kV Tecnai G$^2$ Polara (FEI) with a 100 µm objective aperture. All data were collected using a post-column quantum energy filter (Gatan) with a 20 eV slit and a K2 Summit direct detector (Gatan) operating in counting mode. Dose-fractionated micrographs were recorded in an automated manner using SerialEM (*Mastronarde, 2005*) with a defocus range of –0.8 to –3.0 µm. All datasets were recorded at a nominal magnification of 37,313 corresponding to a pixel size of 1.34 Å/pixel with a total exposure time of 12 s (30 individual frames) and a dose of approximately 1.3 e⁻/Å$^2$/frame. The total electron dose on the specimen level for all datasets was approximately between 32 e⁻/Å$^2$ and 55 e⁻/Å$^2$.

## Cryo-EM image processing

Micrographs from all four datasets were pre-processed in the same manner. Briefly, all individual frames were used for correction of the beam-induced movement using a dose-weighting scheme in RELION's own implementation of the MotionCor2 algorithm available in version 3.0 (*Zivanov et al., 2018*). The CTF parameters were estimated on summed movie frames using CTFFIND4.1 (*Rohou and Grigorieff, 2015*). Low-quality micrographs showing a significant drift, ice contamination or poor CTF estimates were discarded resulting in datasets of 1,125 images of CALHM4 in the presence of Ca$^{2+}$ (dataset 1), 717 images of Ca$^{2+}$-free CALHM4 (dataset 2), 1,059 images of CALHM6 (dataset 3) and 2,065 images of CALHM2 (dataset 4), which were subjected to further data processing in RELION (*Scheres, 2012*). Particles of CALHM4 and CALHM6 were initially picked using the Laplacian-of-Gaussian method and subjected to 2D classification. 2D class averages showing protein features were subsequently used as templates for more accurate auto-picking as well as input for generating an initial 3D model. From dataset one, 422,281 particles were extracted with a box size of 234 pixels, down-scaled three times and subjected to 2D classification. Having discarded false positives and particles of poor quality, the dataset was reduced to 201,782 particles. Two rounds of non-symmetrized 3D classification using the 60 Å low-pass filtered initial 3D model as a reference allowed to isolate two populations of homogenous particles representing dihedrally-related decameric and undecameric assemblies. These two subsets were refined separately with either D10 or D11 symmetry imposed followed by unbinning to an original pixel size and iterative rounds of 3D refinement, per-particle CTF correction and Bayesian polishing (*Zivanov et al., 2018*; *Zivanov et al., 2019*). Extra 3D classification without angular alignment showed increased flexibility within the dimerization interface of interacting intracellular regions in the decameric assembly while no such flexibility was observed for the undecameric population. In order to improve the resolution of the reconstructions, localized reconstruction was performed. For this purpose, the signal corresponding to the detergent belt and to one dihedrally-related monomer was subtracted from each particle followed by auto-refinement of merged in silico modified particles in the presence of a soft mask around the protein density with either C10 or C11 symmetry imposed. The final map of the decameric and undecameric assembly was improved to 4.07 Å and 3.92 Å, respectively. The maps were sharpened using isotropic b-factors of –200 Å$^2$ and –177 Å$^2$, respectively. Datasets of Ca$^{2+}$-free CALHM4 and Ca$^{2+}$-CALHM6 were processed in a similar manner. Briefly, from dataset two, 576,841 particles were extracted and cleaned by 2D classification. The pool, reduced to 97,978 particles, was subjected to non-symmetrized 3D classification that also yielded two populations of dihedrally-related decameric and undecameric assemblies. The final map of the decameric assembly at 4.07 Å and of the undecameric assembly at 3.69 Å was sharpened using isotropic b-factors of –169 Å$^2$ and –126 Å$^2$, respectively. From dataset three, 216,859 particles were extracted with a box size of 200 pixels and reduced to 201,761 particles after 2D classification. Non-symmetrized 3D classification also revealed decameric and undecameric populations, although no dihedrally-symmetrized dimers, as in case of CALHM4 in the presence and absence of Ca$^{2+}$, were observed. The final auto-refined map of the decameric assembly at 4.39 Å and of the undecameric assembly at 6.23 Å was sharpened using isotropic b-factors of –259 Å$^2$ and –435 Å$^2$, respectively. In all cases, resolution was estimated in the presence of a soft solvent mask and based on the gold standard Fourier Shell Correlation (FSC) 0.143 criterion (*Chen et al., 2013*; *Rosenthal and Henderson, 2003*; *Scheres, 2012*; *Scheres and Chen, 2012*). High-resolution noise substitution was applied to correct FSC curves for the effect of soft masking in real space (*Chen et al., 2013*). The local resolution was estimated using RELION (*Zivanov et al., 2018*).

Visual inspection of micrographs in dataset 4 (CALHM2) during pre-processing hinted at preferential orientation of particles. 2D class averages generated from the particles picked with the Laplacian-of-Gaussian method showed primarily views from the extracellular side with a small percentage of side or tilted views, similar to those found in CALHM4 datasets (dihedrally-related dimers) and the CALHM6 dataset (monomers). In order to recover projections at orientations other than from the extracellular side, representative 2D class averages of CALHM4 and 6 were combined and used as 2D templates for auto-picking. The pool of 417,612 particles was subjected to two rounds of 2D classification, after which the dataset was reduced to 71,555 particles. 2D class averages showed that the majority of the remaining particles comprised a view from the extracellular side with only a small fraction representing other orientations. As a consequence of this preferential orientation, projections from other angles were underpopulated and did not show high-resolution features. In other to separate monomeric and dimeric populations, non-symmetrized 3D classification was performed using low-pass filtered reconstructions of CALHM4 and CALHM6. However, both monomeric and dimeric 3D reconstructions of CALHM2 suffered from missing views and therefore were not able to converge to a reliable model during 3D auto-refinement. Additional 2D classification on particles classified either as CALHM2 monomers or dimers showed significant amount of remaining heterogeneity in form of undecameric and dodecameric assemblies, which together with the preferential orientation impeded obtaining a high-resolution reconstruction from this dataset.

## Model building and refinement

The models of CALHM4 and CALHM6 were built in Coot (*Emsley and Cowtan, 2004*). CALHM4 was built de novo into the cryo-EM density of the dimer of undecameric channels at 3.82 Å. The slightly better resolved cryo-EM density of the unpaired undecameric channel at 3.69 Å obtained by localized reconstruction and a map blurred in Coot with a *b*-factor of 50 aided map interpretation. CALHM6 was built using the CALHM4 structure as a reference with the aid of modified cryo-EM maps of CALHM6 which were either low-pass filtered to 6 Å, blurred in Coot with a *b*-factor of 200 or sharpened in Coot with a *b*-factor of −50. The cryo-EM density of CALHM4 was of sufficiently high resolution to unambiguously assign residues 4–83 and 94–280. The cryo-EM density of CALHM6 allowed us to assign residues 20–82 and 94–282. The atomic models were improved iteratively by cycles of real-space refinement in PHENIX (*Adams et al., 2002*) with secondary structure and 22-fold (for CALHM4) and 10-fold (for CALHM6) NCS constraints applied followed by manual corrections in Coot. Validation of the models was performed in PHENIX. Surfaces were calculated with MSMS (*Sanner et al., 1996*). Figures and videos containing molecular structures and densities were prepared with DINO (http://www.dino3d.org), PyMOL (*DeLano, 2002*), Chimera (*Pettersen et al., 2004*) and ChimeraX (*Goddard et al., 2018*).

## Analysis of the density inside the pore

For analysis of the density inside the pore, non-symmetrized final 3D reconstructions of CALHM4 and CALHM6 were opened as a stack in Fiji (*Schindelin et al., 2012*), where the mean density of the area inside the pore was quantified along the slices of each 3D reconstruction. The size of the measured area was chosen based on the 3D mask generated from the atomic models of CALHM4 and 6. In both cases, the area of 6 × 6 pixels centered around the pore axis was located outside the 3D mask ensuring that the procedure did not include density of CALHM4 or 6.

The 'experimental' reference profile was generated based on the electron density map of the MPEG-1 protein bound to a lipidic vesicle (EMD-20622) (*Pang et al., 2019*). The electron density corresponding to the protein was selected and subtracted in Chimera using the structure of MPEG-1 (PDBID 6U2W). Due to the membrane deformation, the central region of the bilayer was excluded from the measurement. The 'simulation' reference profile was generated using the atomistic model of a membrane obtained from the MemProtMD database (PDBID 2N5S) (*Mineev et al., 2015*; *Newport et al., 2019*). All atoms corresponding to protein, ions and water molecules were removed from the model and the remaining lipids were used to generate an electron density map with 1.34 Å pixel spacing at 6 Å resolution using Chimera (*Pettersen et al., 2004*). Electron density profiles were generated by measuring a mean pixel intensity of selected regions on map slices along the Z-axis using the Fiji software (*Schindelin et al., 2012*).

## Two-electrode voltage-clamp recording

For preparation of cRNA coding for CALHM paralogs, *CALHM*-pTLNX DNA constructs were linearized with the FastDigest MluI restriction enzyme (ThermoFisher) purified, transcribed in vitro with the mMessage mMachine kit (Ambion) and purified with the RNeasy kit (Qiagen). The obtained cRNAs were either used immediately or aliquoted and stored at -20℃. Defolliculated *X. laevis* oocytes obtained from Ecocyte Bioscience, were injected with mixtures containing either 1 ng of *CALHM1* cRNA or 5 ng of *CALHM 2, 4 and 6* cRNA and 10 ng of *X. laevis* connexin-38 antisense oligonucleotide (Cx38 ASO) to inhibit endogenous Cx38 currents (*Bahima et al., 2006*). Oocytes used as negative control (neg.) were injected with 10 ng of Cx38 ASO only. For protein expression, oocytes were kept in ND69 solution (96 mM NaCl, 2 mM KCl, 1.8 mM $CaCl_2$, 1 mM $MgCl_2$, 2.5 mM Na-pyruvate, 5 mM HEPES and 50 µg/ml gentamicin, pH 7.5) at 16℃. Two-electrode voltage-clamp (TEVC) measurements were performed 48–60 hr after RNA injections at 20℃. TEVC data were recorded on an OC-725B amplifier (Warner Instrument Corp.) Data recorded at 5 kHz and filtered at 1 kHz were and digitized using a Digidata interface board (1322A or 1440A, Axon Instruments) and analyzed with pCLAMP 10.3 software (Molecular Devices, Sunnyvale, CA). Microelectrodes with a resistance of 1–4 $M\Omega$ were filled with 3 M KCl. A VC-8 valve controller (Warner Instruments) was used for perfusion of different $Ca^{2+}$ concentrations. Prior to recording, oocytes were additionally injected with a 50 nl of a mixture containing 20 mM BAPTA and 10 mM $Ca^{2+}$ solution to minimize activation of $Ca^{2+}$-activated $Cl^-$ currents. High $Ca^{2+}$ bath solutions contained,100 mM $Na^+$, 5.4 mM $K^+$, 95 mM $Cl^-$, 1 mM $Mg^{2+}$, 3 mM $Ca^{2+}$ and 10 mM HEPES, pH 7.2. Divalent cation-free solutions contained 0.5 mM EGTA and 0.5 mM EDTA instead of divalent cations. Intermediate $Ca^{2+}$ concentrations were prepared from both stocks by mixing solution according to the volume calculated with WEBMAXC calculator.

## Cell surface biotinylation

Surface biotinylation of proteins expressed in *X. laevis* oocytes was performed using the Pierce Cell Surface Protein Isolation kit. Oocytes (20-50), injected with *CALHM* cRNAs and incubated for 40–60 hr, were washed three times with ND96 solution, transferred to a white 6-well plate (NUNC) and biotinylated by 30 min incubation in 4 ml ND96 solution supplemented with 0.5 mg $ml^{-1}$ EZ-link sulfo-NHS-SS biotin. After incubation, the biotinylation reaction was stopped by addition of quenching solution and oocytes were washed several times with ND96 solution to remove residual reagents. For protein extraction, oocytes were incubated in lysis buffer (25 mM HEPES, 150 mM NaCl, 0.5 mM $CaCl_2$, 1% LMNG, 2 mM $MgCl_2$, protease inhibitors, RNase, DNase, pH 7.6) for 1 hr at 4℃ and centrifuged at 10,000 g for 15 min. The supernatants were collected and incubated with NeutrAvidin agarose slurry for 1 hr at room temperature under constant mixing. After this step, agarose beads binding the biotinylated proteins were washed with wash solution (10 mM HEPES, 150 mM NaCl, 0.0058% GDN, 2 mM $CaCl_2$, protease inhibitors, RNase, DNase). Subsequently, the unbound material was discarded and biotinylated proteins were incubated with SDS-Page sample buffer containing 50 mM DTT for 1 hr at RT. Eluted protein fraction was obtained by centrifugation at 1,000 g for 2 min at RT. Membrane protein samples were stored at −20℃. For western blot analysis, samples were loaded on a 4–20% SDS-polyacrylamide gel. After electrophoretic separation, the proteins were transferred to a polyvinylidene fluoride membrane by a semi-dry blotting procedure. The membranes were first blocked at room temperature for 2 hr with 5% non-fat milk in TBS-T buffer (50 mM Tris, 150 mM NaCl, 0.075% Tween20, pH 7.5) and then incubated with respective anti-CALHM primary antibodies overnight at 4℃. To remove unbound primary antibodies, the membranes were washed with TBS-T buffer and subsequently blotted with goat anti-rabbit-HRP conjugated secondary antibody for 2 hr at 4℃. The membranes were washed again with TBS-T buffer and chemiluminescent signals were developed with the Amersham ECL Prime Western Blotting Detection kit.

## Statistics and reproducibility

Paired 2-way ANOVA with Sidak's multiple comparisons test was applied to detect differences in placental mRNA levels of CALHM isoforms between undifferentiated cytotrophoblast and differentiated syncytiotrophoblast cells. A p-value<0.05 was considered as statistically significant. Statistical comparisons were performed using GraphPad Prism (GraphPad). Electrophysiology data were repeated multiple times with different batches of cRNA and *X. laevis* oocytes with very similar

results. Conclusions of experiments were not changed upon inclusion of further data. In all cases, leaky oocytes were discarded. For $Ca^{2+}$ concentration-response analysis using TEVC methods, statistical significance was determined by analysis of variance and by Student's t test. A $p$-value$<0.05$ was considered statistically significant. The number of independent experimental repetitions is represented by $n$.

## Accession codes

The cryo-EM density maps of CALHM4 in absence of $Ca^{2+}$ and CALHM4 and CALHM6 in presence of $Ca^{2+}$ have been deposited in the Electron Microscopy Data Bank under following ID codes: EMD-10917, EMD-10919, EMD-10920, EMD-10921, EMD-10924 and EMD-10925. The coordinates of the corresponding atomic models of CALHM4 and CALHM6 have been deposited in the Protein Data Bank under ID codes 6YTK, 6YTL, 6YTO, 6YTQ, 6YTV and 6YTX.

## Acknowledgements

This research was supported by a grant from the Swiss National Science Foundation (No. 31003A_163421) to RD. CA acknowledges support from the Swiss National Science Foundation through the National Centre of Competence in Research TransCure and from the Stiftung Lindenhof Bern. We thank O Medalia and M Eibauer, the Center for Microscopy and Image Analysis (ZMB) of the University of Zurich, and the Mäxi foundation for the access to electron microscopes, S Klauser and S Rast for their help in establishing the computer infrastructure. We are grateful to R Karahoda and S Shahnawaz for their expert help with qPCR and immunoblotting. The authors want to express their gratitude to the patients, physicians and midwives of the Lindenhofgruppe Bern, who participated in this study. Special thanks also to R Moser, Lindenhofgruppe Bern, for coordinating the placenta sampling process. Lipid analysis was performed with the help of the FGCZ of UZH/ETH Zurich. The support of Sebastian Streb and Endre Lacko is acknowledged. All members of the Dutzler lab are acknowledged for their help at various stages of the project.

## Additional information

### Funding

| Funder | Grant reference number | Author |
|---|---|---|
| Schweizerischer Nationalfonds zur Förderung der Wissenschaftlichen Forschung | 31003A_163421 | Raimund Dutzler |
| Stiftung Lindenhof Bern | | Christiane Albrecht |
| Schweizerischer Nationalfonds zur Förderung der Wissenschaftlichen Forschung | NCCR TransCure | Christiane Albrecht |

The funders had no role in study design, data collection and interpretation, or the decision to submit the work for publication.

### Author contributions

Katarzyna Drożdżyk, Conceptualization, Data curation, Formal analysis, Validation, Investigation, Visualization, Methodology, Writing - original draft, Writing - review and editing, Generated expression constructs, purified proteins and assisted with structure determination and functional experiments; Marta Sawicka, Conceptualization, Data curation, Formal analysis, Validation, Investigation, Visualization, Methodology, Writing - original draft, Writing - review and editing, Prepared the samples for cryo-EM, collected EM data and proceeded with structure determination; Maria-Isabel Bahamonde-Santos, Conceptualization, Data curation, Formal analysis, Validation, Investigation, Visualization, Methodology, Writing - original draft, Writing - review and editing, Recorded and analyzed electrophysiology data; Zaugg Jonas, Conceptualization, Data curation, Formal analysis, Validation, Investigation, Visualization, Methodology, Writing - original draft, Writing - review and editing, Carried out expression analysis in the placenta; Dawid Deneka, Investigation, Methodology,

Writing - review and editing, Generated and characterized initial constructs; Christiane Albrecht, Conceptualization, Data curation, Formal analysis, Supervision, Funding acquisition, Validation, Visualization, Writing - original draft, Project administration, Writing - review and editing, Carried out expression analysis in the placenta; Raimund Dutzler, Conceptualization, Data curation, Formal analysis, Supervision, Funding acquisition, Validation, Visualization, Writing - original draft, Project administration, Writing - review and editing

## Author ORCIDs
Katarzyna Drożdżyk ⓘ https://orcid.org/0000-0001-6288-4735
Marta Sawicka ⓘ https://orcid.org/0000-0003-4589-4290
Raimund Dutzler ⓘ https://orcid.org/0000-0002-2193-6129

## Ethics
Human subjects: Human placental tissues were collected from the Division of Obstetrics and Gynecology, Lindenhofgruppe Bern, Switzerland, under approval by the ethical commission of the Canton of Bern (approval No Basec 2016-00250). Written informed consent was obtained from all participants.

## Decision letter and Author response
Decision letter https://doi.org/10.7554/eLife.55853.sa1
Author response https://doi.org/10.7554/eLife.55853.sa2

# Additional files
## Supplementary files
• Supplementary file 1. Key resources table.

• Transparent reporting form

## Data availability
Coordinates of the atomic models were deposited with the PDB and cryo-EM densities were deposited with the Electron Microscopy databank.

The following datasets were generated:

| Author(s) | Year | Dataset title | Dataset URL | Database and Identifier |
|---|---|---|---|---|
| Sawicka M, Drozdzyk K, Dutzler R | 2020 | Cryo-EM structure of a dimer of decameric human CALHM4 in the absence of Ca2+ | https://www.rcsb.org/structure/6YTK | RCSB Protein Data Bank, 6YTK |
| Sawicka M, Drozdzyk K, Dutzler R | 2020 | Cryo-EM structure of a dimer of undecameric human CALHM4 in the absence of Ca2+ | https://www.rcsb.org/structure/6YTL | RCSB Protein Data Bank, 6YTL |
| Sawicka M, Drozdzyk K, Dutzler R | 2020 | Cryo-EM structure of a dimer of decameric human CALHM4 in the presence of Ca2+ | https://www.rcsb.org/structure/6YTO | RCSB Protein Data Bank, 6YTO |
| Sawicka M, Drozdzyk K, Dutzler R | 2020 | Cryo-EM structure of a dimer of undecameric human CALHM4 in the presence of Ca2+ | https://www.rcsb.org/structure/6YTQ | RCSB Protein Data Bank, 6YTQ |
| Sawicka M, Drozdzyk K, Dutzler R | 2020 | Cryo-EM structure of decameric human CALHM6 in the presence of Ca2+ | https://www.rcsb.org/structure/6YTV | RCSB Protein Data Bank, 6YTV |
| Sawicka M, Drozdzyk K, Dutzler R | 2020 | Cryo-EM structure of undecameric human CALHM6 in the presence of Ca2+ | https://www.rcsb.org/structure/6YTX | RCSB Protein Data Bank, 6YTX |
| Sawicka M, Drozdzyk K, Dutzler R | 2020 | Cryo-EM structure of a dimer of decameric human CALHM4 in the | https://www.ebi.ac.uk/pdbe/entry/emdb/EMD- | Electron Microscopy Data Bank, 10917 |

| | | | absence of Ca2+ | 10917 | |
|---|---|---|---|---|---|
| Sawicka M, Drozd-zyk K, Dutzler R | 2020 | Cryo-EM structure of a dimer of undecameric human CALHM4 in the absence of Ca2+ | https://www.ebi.ac.uk/pdbe/entry/emdb/EMD-10919 | Electron Microscopy Data Bank, 10919 |
| Sawicka M, Drozd-zyk K, Dutzler R | 2020 | Cryo-EM structure of a dimer of decameric human CALHM4 in the presence of Ca2+ | https://www.ebi.ac.uk/pdbe/entry/emdb/EMD-10920 | Electron Microscopy Data Bank, 10920 |
| Sawicka M, Drozd-zyk K, Dutzler R | 2020 | Cryo-EM structure of a dimer of undecameric human CALHM4 in the presence of Ca2+ | https://www.ebi.ac.uk/pdbe/entry/emdb/EMD-10921 | Electron Microscopy Data Bank, 10921 |
| Sawicka M, Drozd-zyk K, Dutzler R | 2020 | Cryo-EM structure of decameric human CALHM6 in the presence of Ca2+ | https://www.ebi.ac.uk/pdbe/entry/emdb/EMD-10924 | Electron Microscopy Data Bank, 10924 |
| Sawicka M, Drozd-zyk K, Dutzler R | 2020 | Cryo-EM structure of undecameric human CALHM6 in the presence of Ca2+ | https://www.ebi.ac.uk/pdbe/entry/emdb/EMD-10925 | Electron Microscopy Data Bank, 10925 |

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
