## [Decision Letter]

**Acceptance summary:**

The transport of substances across the placenta is essential for the development of the fetus. In this manuscript, Katarzyna Drozdzyk, Marta Sawicka and their colleagues identified three subtypes of large-pore calcium homeostasis modulator (CALHM) channels in human placenta based on transcript analysis and solved near-atomic resolution cryo-EM structures of CALHM4 and CALHM6, revealing similar overall architectures, with interesting differences in the TM1 helix that lines the central permeation pathway. In CALHM4, TM1 packs against the rest of the channel, creating a wide, cylindrical pore that remarkably appears to be filled with lipids, which they also measure using mass spectrometry. In CALHM6, the density for TM1 is less well-resolved, but adopts a bent conformation extending towards the central axis of the channel to give the pore a conical shape. The authors also clearly demonstrate that CALHM channels can be assembled from different numbers of subunits, with both CALHM4 and CALHM6 channels showing both decameric and undecameric forms, suggesting that the energy penalty for inserting additional subunits into the oligomeric channel is low. These new structures raise fascinating possibilities for how CALHM channels might open and close to enable transport of substrates and ions, including alternating between conical and cylindrical pores, with lipids filling the pore in the closed state and reorganizing to provide a permeation pathway in the open state. These interesting new results are presented with clarity, making provocative and important contributions to our understanding of this intriguing family of large-pore CALHM channels.

**Decision letter after peer review:**

Thank you for submitting your article "Cryo-EM structures and functional properties of CALHM channels of the human placenta" for consideration by *eLife*. Your article has been reviewed by three peer reviewers, and the evaluation has been overseen by Kenton Swartz as the Senior and Reviewing Editor. The following individual involved in review of your submission has agreed to reveal their identity: Wei Lü (Reviewer #3). The reviewers have discussed the reviews with one another and the Reviewing Editor has drafted this decision to help you prepare a revised submission.

Summary:

In this report by Drozdzyk and colleagues, three members of the CALHM family (CALHM2, CALHM4 and CALHM6) are identified as being highly abundant in human placenta tissue based on transcript analysis. However, no currents could be detected from these three paralogs by two-electrode voltage clamp in *Xenopus oocytes*, while cells expressing CALHM1 displayed Ca^2+^ and voltage-dependent currents. FSEC analysis of transiently expressed CALHM2, CALHM4 and CALHM6 revealed the presence of large oligomeric assemblies of varying size, which was supported by single-particle cryo-EM analysis of purified proteins. Single-particle cryo-EM density maps of CALHM4 and CALHM6 at 3.9 Å and 4.5 Å, respectively, revealed similar overall architectures with the exception of TM1. In CALHM4, TM1 packs against the rest of the channel, creating a wide, cylindrical pore that appears to be filled with lipids. In CALHM6, the density TM1 is poorly resolved, but in lower resolution map can be seen adopting a bent conformation extending towards the central axis of the channel. The lower resolution precludes assessment of the functional state of CALHM6, but the presence of a continuous bilayer-like structure in CALHM4 suggests that it is represents a non-conductive state. Based on these two structures, the authors propose a model whereby exchanging between the two states may be associated with channel gating due to the displacement of the lipid bilayer. The data are nicely presented and contribute to the growing understanding of this family of six putative ion channels. Many questions remain, but the availability of these structures will now enable testing of specific hypotheses, most notably whether these so-called "large" CALHM proteins are actually ion channels.

Essential revisions:

1) The authors observed unresolved densities within the cylindrical pore of CALHM4, and found the distribution of these densities approximately matches the one of lipid bilayer density derived from MD simulation. Therefore, they believe these densities represent lipid/detergent molecules. While the authors could be right that these densities are indeed from lipid/detergent molecules, they might want to turn down this conclusion as the evidence in the present manuscript not conclusive. Moreover, such a distribution of unresolved densities is not present in the CALHM6 (this study) and CALHM2 structures (Choi et al., 2019), both have a conical pore; while the former has been discussed in the manuscript, the latter should also be clearly mentioned. If the authors want to present the idea of lipids filling the pore, we would highly recommend doing mass spectroscopic analysis of CALHM4 and CALHM6 to identify these densities and their abundance to better understand the differences between CALHM4 and CALHM6.

2) The authors proposed the lipid-gated mechanism based on their observation of unresolved densities within the pore of CALHM4 that may represent lipid or detergent molecules. We have several concerns regarding this mechanism for the following reasons (Figure 8, right panels).

– The S1 helix tightly interacts with S3 in the resting closed state in which the cylindrical-shaped pore is filled by lipid molecules (Discussion, second paragraph). Where does the energy come from to break such tight interactions, and to push out the lipid molecules to open the pore?

– After the lipids are pushed out by the S1 helix, how do they leave the pore and where do they go?

– When the channel transits from open to closed state, how the lipids are recruited to fill the pore? What are the possible mechanism and pathways for the lipid molecules to re-enter the pore?

– The authors hypothesized the unresolved densities within the pore are lipids with one of the reasons being that the S1 helix contains mostly aliphatic residues, which makes the pore highly hydrophobic. If this is the case, the conformation of the S1 helix in the open state (Figure 8 lower right) seems energetically very unfavorable because the majority of the S1 helix becomes solvent exposed.

– If the authors believe the space between the S1 and S3 helices are filled by lipid molecules in the conical pore of the open state (Figure 8 lower right), why should and could the lipid molecules be removed from other places of the pore, particularly in the extracellular half of the pore?

3) TM1 of CALHM6 is poorly ordered in the structure and resolvable only in a map low-pass filtered at 6 Å. To model TM, the authors relied on the close homology with CALHM4. Because the gating mechanism proposed by the authors relies on differences between the conformations of TM1 between CALHM4 and CALHM6, focused refinements of individual and several adjacent subunits, both with or without signal subtraction should be employed to better understand the conformational diversity of TM1 and its effects on the pore. If focused refinements do not help resolve the density, it would be better if the side chains are excluded from the model.

4) The authors suggested that, based on 2D class averages in top/down views, human CALHM2 assembles as both 11-mer and 12-mer. This is different from recent two published reports that human CALHM2 is a 11-mer. They should discuss potential reasons that may cause such a difference.

5) The CALHM4 assembles as both 10-mer and 11-mer, which is a very interesting observation. But why do two different stoichiometries co-exist for the same protein? Are both stoichiometries physiologically relevant?

6) CALHM4 assembles as 10-mer and 11-mer, yet with the same CTH helix in both structures. This observation is somewhat different from the hypothesis by Syrjanen et al., 2020 that CTH determines the oligomeric state of CALHMs, and should be discussed in the manuscript.

Other suggestions:

1) If the authors have access to tissue sections, it would be helpful to employ immunohistochemistry using the antibodies from Figure 2 to characterize the localization of the various CALHM proteins in the placenta, especially as their molecular function is poorly understood.

2) A comparison between the structures of the non-conducting CALHM4 and CALHM6 and the conducting CALHM1 pores in terms of size and hydrophobicity would be helpful in understanding the origin of the proposed lipid bilayers in CALHM4 and CALHM6 and their potential role in gating, as is proposed in Figure 8.

3) In the subsection “Biochemical characterization and structure determination of CALHM2, 4, and 6”, the authors claim that CALHM2 forms both cylindrical pore seen in CALHM4 structure and the canonical pore seen in the CALHM6 structure? While both paired and unpaired structures can be readily distinguished from the averages shown in Figure 3—figure supplement 6B, it is unclear how the conformation of the pore is identified from inspection of the 2D averages. It would be helpful in Figure 3—figure supplement 6B to identify 2D class averages that display 11 subunits and those that display 12. It is difficult from the images shown to distinguish the class averages.

4) The authors have included an ATP molecule in the center of the pore in Figures 4 and 5. This may be confusing to readers as no ATP molecules were resolved in the cryo-EM density map and ATP conductance has not been demonstrated for CALHM4. It would be better left for a figure supplement, where it can be clearly defined as a model and not part of the structure.

---

## [Author Response]

Essential revisions:1) The authors observed unresolved densities within the cylindrical pore of CALHM4, and found the distribution of these densities approximately matches the one of lipid bilayer density derived from MD simulation. Therefore, they believe these densities represent lipid/detergent molecules. While the authors could be right that these densities are indeed from lipid/detergent molecules, they might want to turn down this conclusion as the evidence in the present manuscript not conclusive. Moreover, such a distribution of unresolved densities is not present in the CALHM6 (this study) and CALHM2 structures (Choi et al., 2019), both have a conical pore; while the former has been discussed in the manuscript, the latter should also be clearly mentioned. If the authors want to present the idea of lipids filling the pore, we would highly recommend doing mass spectroscopic analysis of CALHM4 and CALHM6 to identify these densities and their abundance to better understand the differences between CALHM4 and CALHM6.

In our revised manuscript, we have provided a lipid analysis by mass spectrometry, which demonstrates the presence of membrane lipids in our CALHM4 samples. For this experiment, we have purified CALHM4 in the same way as for cryo-EM studies, extracted co-purified lipids in chloroform and analyzed them by liquid chromatography and mass spectrometry. In these samples, we were able to detect several compounds that are not present in the equivalent extraction of the buffer and that can be assigned with confidence to phospholipids and their fragments that are constituents of the HEK cell membranes. The data is presented as Figure 6—figure supplement 2. The chemical detection of lipids further strengthens the possibility that the observed bilayer-like densities in the CALHM4 structure might indeed originate from lipids. After careful analysis, we also find very similar density in the Ca^2+^-free data of CALHM4, which was collected from an independent protein preparation (and which is now described in additional panels of Figure 6—figure supplement 1). Remarkably, a very similar density distribution is also found in 2D projections of dimers of CALHM2 channels shown in Figure 3C (top), which we think also resides in the same cylindrical pore conformation. We also want to emphasize that we do not claim that no lipids would be associated with the conical pore conformation in CALHM6, although we find a clear different distribution of residual density in this channel conformation. In the CALHM6 data, the density assigned to the potential outer leaflet of the membrane is absent, whereas there is density at the position of the inner leaflet. Lipids in this part might thus be less well ordered but they still could be present. We thus refrained from the equivalent analysis of lipids in CALHM6. We also would like to emphasize that the lipid analysis is demanding and has required as much protein as used for structure determination and that we can at the moment not provide additional data in light of the shutdown of research facilities at UZH related to the coronavirus outbreak.

The observation of extra density present inside the pore of CALHM4 is in general agreement with equivalent density observed inside the pore of human CALHM2 (Syrjanen et al., 2020), which also displays a cylindrical pore conformation (although in the displayed map this density is fragmented as a consequence of map sharpening). In their publication, these observations were further supported by molecular dynamics simulations that showed that the CALHM2 undecamer but not the CALHM1 octamer would be able to accommodate lipids in a bilayer-like state. In agreement with our CALHM6 data, no such density is found in the conical conformation of CALHM2 (Choi et al., 2019) which we now also mention in the manuscript.

Changes in the manuscript:

“We thus analyzed the composition of small molecules that are co-purified with CALHM4 by mass spectrometry and were able to detect phospholipids that are commonly found in the membranes of HEK cells (Figure 6—figure supplement 2).”

Addition of Figure 6—figure supplement 2.

Addition of lipid analysis in the Materials and methods.

“A comparable distribution of pore density is found in the structure of CALHM2 in complex with ruthenium red, which resides in a similar conical pore conformation (Choi et al., 2019).”

2) The authors proposed the lipid-gated mechanism based on their observation of unresolved densities within the pore of CALHM4 that may represent lipid or detergent molecules. We have several concerns regarding this mechanism for the following reasons (Figure 8, right panels).– The S1 helix tightly interacts with S3 in the resting closed state in which the cylindrical-shaped pore is filled by lipid molecules (Discussion, second paragraph). Where does the energy come from to break such tight interactions, and to push out the lipid molecules to open the pore?

The open-probability of a channel depends on the energy differences between open and closed conformations, which are populated according to a Boltzmann distribution. During gating, the energy difference between states alters. Depending on the class of channels, opening can be triggered in response to changes in the transmembrane voltage, the binding of agonists or dissociation of antagonists. Although both voltage and Ca^2+^ removal influence the open probability of CALHM1, the activating stimulus of CALHM2, 4 and 6 is at this stage unknown.

– After the lipids are pushed out by the S1 helix, how do they leave the pore and where do they go?– When the channel transits from open to closed state, how the lipids are recruited to fill the pore? What are the possible mechanism and pathways for the lipid molecules to re-enter the pore?

The exact trajectory of lipids during structural changes is still unclear, particularly since, due to the tight interactions between subunits, it is not obvious how lipids might diffuse into and out of the pore region. Currently, we cannot exclude that this might be facilitated by a transient lateral opening of the channel. Alternatively, it is possible that the conformational rearrangement from a cylindrical to a conical conformation would not displace lipids but instead cause their redistribution, which would affect the bilayer-like structure and its barrier function.

– The authors hypothesized the unresolved densities within the pore are lipids with one of the reasons being that the S1 helix contains mostly aliphatic residues, which makes the pore highly hydrophobic. If this is the case, the conformation of the S1 helix in the open state (Figure 8 lower right) seems energetically very unfavorable because the majority of the S1 helix becomes solvent exposed.

This is not necessarily the case, since TM1 rotates during the rearrangement thereby exposing some hydrophilic residues to the lining of the cylindrical pore (see Figure 7F, G, Video 4). It should also be emphasized that parts of the pore lining at the wide entrance of different ion channels can be hydrophobic (as observed in K-channels, pentameric ligand-gated ion channels, bestrophins and volume-regulated chloride channels to name few of them).

– If the authors believe the space between the S1 and S3 helices are filled by lipid molecules in the conical pore of the open state (Figure 8 lower right), why should and could the lipid molecules be removed from other places of the pore, particularly in the extracellular half of the pore?

We could envision a redistribution of lipids upon conformational changes. Although the presence of a lipid bilayer inside the pore of a large channel is an intriguing possibility, it is not clear whether it would have the same energetic properties in such confined environment as in the bulk of the membrane. The energetics for redistribution could thus be smaller than in a larger membrane.

Altogether, the reviewer has raised valid questions, which we can at this stage not fully explain. We want to emphasize that the potential regulation of the pore by lipids is at this stage speculative and it is put forward as one of two alternative mechanisms how conduction in a large channel could be regulated. Whereas the presence of lipids inside the pore could explain why we did not observe conduction in placental CALHM channels, despite their apparent large pore diameter, it is at this stage not clear whether such regulation might at all be of physiological relevance. Potentially, the bilayers inside the pore could be an artefact of overexpression and they might form in the absence of a regulatory protein, which would be present in a physiological context. It should also be mentioned that the presence of lipids inside the pore of large channels is a novel but recurrent theme which has also been proposed for connexins and VRAC channels.

Changes in the manuscript:

“Both models are at this stage hypothetical and it is still unclear how lipids in the pore would rearrange during activation.”

3) TM1 of CALHM6 is poorly ordered in the structure and resolvable only in a map low-pass filtered at 6 Å. To model TM, the authors relied on the close homology with CALHM4. Because the gating mechanism proposed by the authors relies on differences between the conformations of TM1 between CALHM4 and CALHM6, focused refinements of individual and several adjacent subunits, both with or without signal subtraction should be employed to better understand the conformational diversity of TM1 and its effects on the pore. If focused refinements do not help resolve the density, it would be better if the side chains are excluded from the model.

By careful reprocessing of the data, we were able to substantially improve the quality of the CALHM6 structure. Although the nominal resolution only improved from 4.5 to 4.4 Å, many features of the map including the region connecting TM1 with TM2 are well-defined and allow an unambiguous interpretation by an atomic model. Due to the high sequence conservation in this region, conformational differences can be described with confidence.

Despite the application of different approaches to improve the resolution of TM1 (summarized below), none of them has led to a better definition of this part of the protein. Our model thus starts with residue 20. The poor definition of TM1 underlines the intrinsic flexibility of the region in the observed conformation, which is expected in the absence of strong interactions with other parts of the structure. However, since the end of TM1 is well-defined, we are confident about the register of residues along the α-helix. The detailed sidechain conformation is in several cases of course arbitrary.

Attempts to improve the density of TM1 included:

– 3D refinement with the symmetry relaxed to either C5, C2 or C1 to detect any asymmetry of the TM1 and NTH between individual protomers. In all cases, TM1 was resolved to the same extend as in C10 map – the resolution estimates were 4.7 Å, 5.15 Å and 6.1 Å, respectively.

– Focused 3D classification with no symmetry imposed on C10-symmetry expanded particles. In order to increase the sensitivity of the alignment, the regularization parameter T was increased to 40. Out of 10 classes, only one including 50% of particles was further refined to 4.6 Å. No improvement in TM1 density was observed.

– Focused 3D classification with no symmetry imposed on particles where the density of other nine protomers was subtracted. In order to increase the sensitivity of the alignment, the regularization parameter T was increased to 40. Particles (83%) belonging to a selected class were reverted to original particles and subjected to 3D refinement. The class was resolved to 4.6 Å and no improvement in TM1 density was observed.

– Extensive 3D classification with either C1 or C10 symmetry applied. In order to determine any possible conformational heterogeneity independent classifications were done with and without particle alignment. In all cases, TM1 was found in the same conformation although the best achieved resolution was 4.6 Å and the reconstruction did not show better resolved features.

– Focused 3D refinement on C10 symmetry expanded particles with and without signal subtraction did not improve the density of TM1.

4) The authors suggested that, based on 2D class averages in top/down views, human CALHM2 assembles as both 11-mer and 12-mer. This is different from recent two published reports that human CALHM2 is a 11-mer. They should discuss potential reasons that may cause such a difference.

We have obtained populations with different oligomeric states in all of our samples including CALHM2. This feature appears to be more pronounced in our study compared to others and it could potentially be a consequence of the different expression host and mode of transfection (i.e. transient transfection in HEK293 cells in our study vs. virus-based transfection in insect cells in other studies). Nevertheless, it should be emphasized that different oligomers for CALHM2 were also observed by other groups. E.g., in the study by Choi et. al, one of the 3D classes of CALHM2 in the presence of RuR shows a dodecameric assembly accounting for 14.2% of the particles (see Choi at al., 2019, Extended Data Figure 4) although this class has not been processed further. The observed heterogeneity probably reflects the low energetic penalty for the incorporation of an additional subunit to channels of large oligomeric state as found in case of CALHM2, 4 and 6. At this stage we do not know whether this is a consequence of heterologous overexpression.

5) The CALHM4 assembles as both 10-mer and 11-mer, which is a very interesting observation. But why do two different stoichiometries co-exist for the same protein? Are both stoichiometries physiologically relevant?

We observed 10-mer and 11-mer assemblies for CALHM4 and CALHM6. In both cases, the two populations are present at ratios of 40% to 60%. Whereas in case of CALHM4, the predominant assembly is the 11-mer, for CALHM6 it is a 10-mer. Currently we do not know whether both assemblies are physiologically relevant, as different oligomerization states might result from overexpression of the proteins, which is controlled by a strong viral promotor. In any case, we conclude that the energy penalty for the incorporation of an additional subunit to a large CALHM channel is low, which is reflected in the similar abundance of both populations. It should be emphasized that we currently understand very little about the physiological role of the three paralogs investigated in this study as it is still unknown whether the proteins form heteromers, are part of a larger complex or whether there are endogenous factors that control the assembly in a native environment. The ultimate resolution of this question will require purification of the respective proteins from native sources.

6) CALHM4 assembles as 10-mer and 11-mer, yet with the same CTH helix in both structures. This observation is somewhat different from the hypothesis by Syrjanen et al., 2020 that CTH determines the oligomeric state of CALHMs, and should be discussed in the manuscript.

The proposal by Syrjanen et al. is interesting in light of experiments provided in their manuscript. In their study, the claim that the oligomerization of CALHM channels would be determined by the cytoplasmic CTH regions and the linker connecting TM4 and CTH is supported by chimeric proteins, where the transfer of the linker and CTH of CALHM2 to CALHM1 changes the oligomerization of the chimeric protein from octamers to undecamers. This observation supports the notion that the C-terminus plays a role in oligomerization and that it contributes to the stability of the octameric assembly found for CALHM1. Different results were obtained from similar CALHM1-CALHM2 chimeras investigated by Demura et al., 2020. In contrast to the study by Syrjanen, their chimera form octa- and nonamers leading to the conclusion that oligomerization would be predominantly driven by inter-subunit interactions in the transmembrane region. Thus, whereas it is likely that the C-terminus would be important for the oligomeric organization of CALHM channels, its contribution might not be as dominant as initially suggested. Particularly, it is unlikely that it would discriminate between the formation of 10- and 11-mers in CALHM4 and 6. Since the CALHM1 structure is not a subject in this study, we prefer not to engage in an extended discussion on the role of the C-terminus on oligomerization.

Other suggestions:1) If the authors have access to tissue sections, it would be helpful to employ immunohistochemistry using the antibodies from Figure 2 to characterize the localization of the various CALHM proteins in the placenta, especially as their molecular function is poorly understood.

We agree that the subcellular localization of the different CALHM paralogs in placental tissue, is one of the critical next steps that will have to be performed to provide insight into the potential role of CALHM channels in the placenta. However, this work relies on highly specific antibodies to be able to come up with meaningful conclusions. We have already started some experiments in this direction, but at the moment found only the CALHM2 antibody to be suitable for such studies. Whereas all three antibodies used in this study are subunit-specific, as judged from Western blots of purified CALHM2, CALHM4 and CALHM6 proteins, the CALHM4 and CALHM6 antibodies also recognize other proteins in tissue lysates as well as isolated placental membrane fractions, which are probably not related to CALHM channels. This is presumably due to the fact that the all antibodies are polyclonal. Evidence for this behavior is found in Figure 2A where the diffuse band between 37 and 50 kDa probably originates from proteins other than CALHM4. We are currently putting large efforts in obtaining better antibodies to clarify the question of tissue localization in a future study.

2) A comparison between the structures of the non-conducting CALHM4 and CALHM6 and the conducting CALHM1 pores in terms of size and hydrophobicity would be helpful in understanding the origin of the proposed lipid bilayers in CALHM4 and CALHM6 and their potential role in gating, as is proposed in Figure 8.

Whereas the sequence alignment of TM1 helices of CALHM1 and CALHM4 pores does not show a strong difference in hydrophobicity of presumable pore-lining residues, detailed differences of the pore region of both proteins cannot be determined based on currently available structures, since TM1 of CALHM1 is not well-resolved in the structure determined by Syrjanen et al., (which is the only CALHM1 structure that is currently available in the PDB). This is different in the equivalent CALHM1 structure described by Demura et al. (although maps and models are in this case not yet available). In latter structure, TM1 is resolved and the N-terminal helix NTH appears to fold back towards TM1, which would further constrict the CALHM1 pore and change its polar character. The position of NTH is different in our structure of CALHM4, where NTH is located horizontal to the membrane. Thus both, the strongly reduced pore diameter due to the lower oligomeric state of CALHM1 and the presence of NTH in the pore lining strongly influences the size and the polar character of the CALHM1 pore, which might prevent the formation of a stable lipid bilayer as suggested by Syrjanen et al. based on molecular dynamics simulations.

3) In the subsection “Biochemical characterization and structure determination of CALHM2, 4, and 6”, the authors claim that CALHM2 forms both cylindrical pore seen in CALHM4 structure and the canonical pore seen in the CALHM6 structure? While both paired and unpaired structures can be readily distinguished from the averages shown in Figure 3—figure supplement 6B, it is unclear how the conformation of the pore is identified from inspection of the 2D averages. It would be helpful in Figure 3—figure supplement 6B to identify 2D class averages that display 11 subunits and those that display 12. It is difficult from the images shown to distinguish the class averages.

Indications for a difference in the pore conformation in distinct populations of CALHM2 channels are found in the 2D classes shown in Figure 3C. The bilayer-like pore density in the paired channel shown on top indicates a cylindrical conformation of the protein, whereas the altered density in the unpaired channel below might correspond to a conical conformation. However, since the evidence is weak compared to the data presented for CALHM4 and CALHM6, we removed the claim that we find both conformations in the data of CALHM2 in the revised manuscript. We also labeled 2D classes of top views showing either 11- or 12-mers in Figure 3—figure supplement 6B.

Changes in the manuscript:

“Although 2D class averages of CALHM2 appear of high quality, 3D classification of this dataset did not yield high-resolution structures. We believe that the strong preferential orientation of CALHM2 particles resulting in a predominance of views along the pore axis combined with sample heterogeneity has limited our data processing workflow to 2D classification”.

4) The authors have included an ATP molecule in the center of the pore in Figures 4 and 5) This may be confusing to readers as no ATP molecules were resolved in the cryo-EM density map and ATP conductance has not been demonstrated for CALHM4. It would be better left for a figure supplement, where it can be clearly defined as a model and not part of the structure.

We have removed the ATP molecules in Figures 4 and 5, but we kept them in Figure 4—figure supplement 1D, E to show the size of this potential substrate in relation to the pore diameter.